

# Detecting irrigation signals from SMAP L3 and L4 soil moisture: A case study in California's Central Valley

Xin Huang[1], Qing He[1], Naota Hanasaki[1,2], Rolf H. Reichle[3], Taikan Oki[1]

[1]Department of Civil Engineering, The University of Tokyo, Tokyo, Japan

[2]Center of Climate Change Adaptation, National Institute for Environmental Studies, Tsukuba, Japan

[3]Global Modeling and Assimilation Office, NASA Goddard Space Flight Center, Greenbelt, MD, USA

*Correspondence to*: Qing He (heqing@g.ecc.u-tokyo.ac.jp) and Xin Huang (xhuangut@gmail.com)

**Abstract:** Recent advances in satellite-based soil moisture observations present a promising opportunity to monitor irrigation dynamics from space and support the refinement of hydrological and land surface model simulations. This study presents an

approach for identifying irrigation signals using data from the Soil Moisture Active and Passive (SMAP) mission, with feasibility demonstrated in Central Valley, California. The approach leverages two SMAP products: the Level 3 Enhanced product, which provides satellite-based soil moisture observations that inherently capture irrigation effects, and the Level 4 data assimilation product, which incorporates only anomalies from SMAP Level 1 brightness temperature data, thereby excluding irrigation effects. The approach is based on the hypothesis that, after correcting for systematic differences not related

to irrigation, the soil moisture difference between the Level 3 and 4 products during the cropping season is primarily attributable to irrigation. This hypothesis is first verified by evaluating soil moisture consistency (i.e., temporal variability and long-term mean values) between the two products. For grid cells that meet this criterion, the mean difference (*MD*) between the two soil moisture products is calculated, separately for the cropping and non-cropping seasons, and then the irrigation signal is identified as the difference in *MD* between the cropping and non-cropping seasons. Validation of the estimated

irrigation signal is made by comparing with two benchmark irrigation maps. The results show reasonable spatial correlations between our estimate and the two benchmark maps, with Pearson's correlation coefficients of 0.66 and 0.50, respectively. Findings demonstrate the potential of using SMAP products to extract irrigation effects in regions that have limited precipitation during the cropping season. Compared to other satellite-based irrigation detection studies, the proposed method requires minimal additional data and avoids additional model tuning beyond the SMAP processing.

**1 Introduction**

Irrigated agriculture plays a critical role in meeting global food demands, contributing approximately 40% of the world's agricultural products (FAO and UN Water, 2021). Despite covering only 2% of the Earth's land area, it accounts for nearly 90% of global water consumption (McDermid et al., 2023; Ritchie and Roser, 2018). However, detailed information on the spatio-

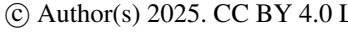

temporal distribution of irrigation water use at the global scale remains largely unavailable. Although some studies have attempted to estimate the irrigation water use (Guillaumot et al., 2022; McDermid et al., 2023; Wada et al., 2013), the absence of continuous, large-scale in-situ measurements hampers both validation and accurate estimation (Hanasaki et al., 2008; Müller Schmied et al., 2021; Sutanudjaja et al., 2018). Currently, only limited regional in-situ measurements are available (Hanasaki et al., 2022), and these datasets provide neither long records nor sufficient spatial coverage for large-scale estimation. This limited validation capability challenges estimation accuracy and underscores the urgent need for the development of large-

scale irrigation monitoring datasets. Yet, factors such as the high cost and limited availability of monitoring equipment and the variability of irrigation practices (See et al., 2023) make it challenging to obtain continuous, large-scale, in-situ irrigation monitoring data at the global scale (Condon et al., 2021; Ozdogan et al., 2010).

In contrast to in-situ measurement networks, satellite observations provide a promising alternative for irrigation monitoring (Dorigo et al., 2021; Zhang et al., 2022). With obvious advantages in extensive spatial coverage, continuous time series, and

compatibility with grid-based hydrological models (Alfieri et al., 2022), satellite observations offer a unique and convenient tool for monitoring key state variables related to irrigation, such as soil moisture and evapotranspiration (Foster et al., 2020; Ihuoma et al., 2021).

Satellite-based irrigation monitoring generally follows one of two main approaches. The first approach is the evapotranspiration-based method, which assimilates remotely sensed evapotranspiration data into hydrological and energy

balance models. This method compares evapotranspiration under conditions without any irrigation inputs to the total evapotranspiration observed, thereby separating evapotranspiration into precipitation-caused and irrigation-caused parts, allowing for the estimation of irrigation water (Gowda et al., 2008; Santos et al., 2008). This method has been successfully applied in some areas such as southern Spain (Droogers et al., 2010), northwest China (Ma et al., 2018), and the Indus basin (Bastiaanssen et al., 2012). However, the retrieval of evapotranspiration products from multispectral sensors is often limited

by weather conditions (Zhang et al., 2016), especially cloud coverage, and by the heavy reliance on hydrological modeling, which demands extensive ancillary data. Therefore, evapotranspiration products are subject to considerable errors and gaps in spatial and temporal coverage.

The second approach for satellite-based irrigation monitoring focuses directly on the core variable that is most relevant to irrigation: soil moisture. Unlike evapotranspiration monitoring, satellite soil moisture observations are primarily derived from

microwave radiometers that operate independently of weather conditions. Since soil emissivity is closely linked to soil moisture, increased moisture results in more pronounced absorption and scattering of microwaves by the soil. Consequently, microwave radiometers are highly sensitive to soil moisture variations (Njoku and Entekhabi, 1996). Major satellite soil



moisture products include those from the Soil Moisture Active Passive (SMAP) mission (Entekhabi et al., 2010) and the Soil

Moisture and Ocean Salinity (SMOS) mission (Barre et al., 2008). The basic principle for this soil moisture-based method is

taking the difference between two soil moisture time series with irrigation effects (usually from satellite products) and without

irrigation effects (usually from model simulations without considering irrigation events) (Brocca et al., 2018).

The key to the soil moisture-based irrigation monitoring approach is to ensure that the time series with and without irrigation

effects are climatologically consistent. Specifically, during the non-cropping season or at non-irrigated grid cells, the model-

simulated soil moisture should be consistent with satellite data in terms of its temporal variability and long-term mean. Several

recent studies have successfully applied this approach but require complex model tuning (Jalilvand et al., 2021; Soylu and

Bras, 2024). Several other studies, however, also demonstrate the possibility to avoid such complex model tuning, although

these approaches may exhibit limitations in maintaining climatological consistency. For example, Zaussinger et al. (2019) used

MERRA-2 reanalysis soil moisture data as the soil moisture without irrigation effects, and calculated the difference between

several satellite soil moisture products such as SMAP, AMSR2, and ASCAT to estimate irrigation water use amount. However,

because the spatiotemporal statistics of MERRA-2 soil moisture are different from any of the satellite products, the identified

irrigation events as well as the estimated irrigation water amount are subject to considerable risk of biases. On the other hand,

in an algorithm where only one satellite product is used, Lawston et al. (2017) compared the soil moisture time series of two

neighboring grid cells with and without irrigation effects, with the irrigation information known a priori, and took the difference

between these two attributed to irrigation. While this method ensures the climatological consistency of the data, the grid cell-

by-grid cell comparisons may make the approach too labor intensive for applications across larger domains.

To address these limitations, the present study attempts to extend the spatial applicability of satellite products to detect

irrigation effects without the need for additional model tuning beyond the standard SMAP processing and without sacrificing

the climatological consistency of the data. To achieve this goal, we take advantage of the different processing levels of SMAP

products, i.e., the Level 3–SMAP Enhanced L3 Radiometer Global and Polar Grid Daily 9 km EASE-Grid Soil Moisture

(SMAP L3_E) (ONeill et al., 2023), and the Level 4–SMAP L4 Global 3-hourly 9 km EASE-Grid Surface and Root Zone Soil

Moisture Geophysical Data (SMAP L4) (Reichle et al., 2022). SMAP L3_E provides near-real-time soil moisture observations

derived directly from microwave radiometer measurements; therefore, the irrigation contributions are embedded in its soil

moisture time series when such practices occur in the real world. In contrast, SMAP L4 assimilates only time series anomaly

information from the SMAP Level 1 brightness temperature ($T_b$) observations into the Catchment Land Surface Model (Koster

et al., 2000), which provides two major advantages here: (1) Since only anomalies of $T_b$ are assimilated, SMAP L4 does not

contain irrigation effects (as further explained in Section 2.1.2); and (2) the SMAP L3_E and L4 algorithms use nearly identical

microwave radiative transfer models, thereby minimizing the systematic differences between the soil moisture estimates from

the two products.

To demonstrate the applicability of our method, we chose the Central Valley in California, USA, one of the most heavily
irrigated regions in the world, as the primary study area, and analyzed SMAP L3_E and L4 data from 2016 to 2022. We also
examined several heavily irrigated regions elsewhere in the contiguous United States (CONUS) to further evaluate the
performance and applicability of the proposed method. Existing studies, including a map of irrigated areas and estimates of
irrigation water use, were used here for validation. Compared to previous studies, the primary innovation of the present study
is in achieving climatological consistency between the datasets and effectively monitoring irrigation effects without additional
model tuning beyond the SMAP processing.

The paper is organized as follows. Section 2 provides a brief introduction to the data and study area. Section 3 details the
methodologies for identifying irrigation signals. Section 4 presents the case study and results, and Sections 5 and 6 offer the
discussion and conclusions, respectively.

## 2 Data and study area

### 2.1 Data

Since its launch in 2015, SMAP has been instrumental in monitoring global soil moisture and freeze-thaw dynamics (Colliander
et al., 2022; Fang et al., 2019; Jonard et al., 2022; Purdy et al., 2018; Melser et al., 2024). The satellite is equipped with an L-
band passive microwave radiometer. L-band microwave emission from the soil is capable of penetrating clouds and modest
amounts of vegetation. Consequently, these $T_b$ observations can be used to retrieve soil moisture estimates by applying a zero-
order ("tau-omega") microwave radiative transfer model (Njoku and Entekhabi, 1996), yielding global estimates of soil
moisture with 2-3 day revisit frequency (depending on latitude). The present study focuses on the SMAP L3_E and L4 products,
which are both provided on the 9 km Equal-Area Scalable Earth (EASE) Grid 2.0. The analysis spans the temporal period
from January 1, 2016 to December 31, 2022. SMAP products are available through the National Aeronautics and Space
Administration (NASA) Earthdata website (https://earthdata.nasa.gov/, last accessed on October 13, 2024).

This study employs SMAP L4, which excludes irrigation effects, as a model-simulated soil moisture reference, and SMAP
L3_E as a satellite-observed soil moisture reference that captures irrigation effects. By analyzing the discrepancies between
the two, we aim to identify potential irrigation signals.

### 2.1.1 SMAP L3_E

Surface soil moisture estimates from the SMAP L3_E product were used. Limited by the soil penetration depth of L-band

microwaves, the nominal detection depth of this product is 5 cm. Nevertheless, surface soil moisture is often representative of

moisture conditions in deeper layers (Feldman et al., 2023). Compared to soil moisture in the root zone or the full soil profile,

surface soil moisture is particularly sensitive to irrigation activities because irrigation directly alters the moisture content within

this layer.

The L3_E product is a daily composite of SMAP Level 2 half-orbit soil moisture data, which are retrieved from SMAP Level

1 $T_b$ data. Although Backus-Gilbert optimal interpolation is applied to post the original 36 km data onto the 9 km EASE Grid

2.0, we note that the relatively coarse effective resolution (closer to 33 km) means that some irrigation features will not be

seen clearly by SMAP. We will further discuss this in Section 4.

SMAP L3_E is categorized into two groups: AM group, collected during the descending pass of the satellite at 06:00 local

solar time, and PM group, collected during the ascending pass at 18:00 local solar time (ONeill et al., 2023). Since the SMAP

soil moisture retrieval algorithm requires ancillary soil temperature inputs, here we use the AM soil moisture retrievals because

in the morning the temperature is typically similar across the soil, vegetation, and near-surface air, and the resulting retrievals

are therefore expected to be of higher quality.

**2.1.2 SMAP L4 and explanation for the absence of irrigation effects**

(1) SMAP L4

SMAP L4 is a model-derived product generated using the Catchment land surface model. It utilizes an ensemble Kalman filter

(EnKF) data assimilation system (Reichle et al., 2017b). This system integrates (temporal) anomalies of SMAP Level 1 $T_b$

observations and provides a comprehensive global soil moisture product that includes the surface layer, the root zone (defined

as 0–1 m depth), and the full soil profile, with complete spatial and temporal coverage (Reichle et al., 2019). Since the SMAP

L3_E product provides only surface soil moisture, we use only surface soil moisture from the SMAP L4 product.

SMAP L4 provides global observations every 3 hours on the 9 km EASE-Grid 2.0. Here, arithmetic averaging was applied to

the 3-hourly data to derive daily values, and the resulting daily SMAP L4 values were cross-masked to match the approximately

2–3-day revisiting frequency of SMAP L3_E to enable a more straightforward comparison.

(2) Explanation for the absence of irrigation effects in SMAP L4

The assimilation process of SMAP L4 incorporates only the anomaly time series from the SMAP Level 1 $T_b$ product. A mean

seasonal cycle of $T_b$ is calculated separately for the Level 1 $T_b$ observations and the corresponding model estimates, where the

mean seasonal cycle is computed as the multi-year mean for a given day of year, with roughly monthly smoothing. Observed

anomalies are then calculated by subtracting the mean seasonal cycle from the original $T_b$ time series of SMAP Level 1

observations; anomalies for the corresponding model estimates are calculated in the same fashion (Reichle et al., 2017a; De

Lannoy and Reichle, 2016). In irrigated regions, where crop area, cultivation type, and irrigation requirements remain relatively

stable over several years, the SMAP Level 1 $T_b$ product reflects the impact of irrigation on soil moisture levels as part of the

monthly seasonal mean, rather than classifying it as an increased anomaly. Consequently, the $T_b$ anomaly time series

assimilated in SMAP L4 inherently excludes irrigation effects, leading to the absence of irrigation effects in the SMAP L4

product.

**2.1.3 Supporting data**

(1) Precipitation

Precipitation is obtained from the Catchment model forcing dataset released with SMAP L4. Over CONUS, this dataset

primarily sources its information from NOAA Climate Prediction Center Unified gauge-based precipitation data. A detailed

description is available in Reichle et al. (2023). The precipitation data can help understand the variability of SMAP L3_E and

L4 and distinguish whether the differences between two soil moisture products are driven by precipitation or by other factors,

such as irrigation.

(2) Map of irrigated area

This study utilized the latest Global Map of Irrigated Area (GMIA) provided by the Food and Agriculture Organization (FAO)

of the United Nations. The GMIA is a static dataset that represents the percentage of irrigated land relative to the total grid cell

area around 2005 (Siebert et al., 2013). It offers global coverage at a 5 arcmin resolution and was resampled to the 9 km EASE-

Grid 2.0 using nearest-neighbor interpolation for comparative analysis.

GMIA is used to identify non-irrigated grid cells and those with varying irrigation fractions. It serves both as the basis for

examining consistency in Section 3.1 and as a validation source for proposed irrigation signals analysis in Section 3.2.

(3) Irrigation water use estimation

To further validate our approach, we compared the derived irrigation signal map with independent irrigation water use

estimations. Currently, large-scale in-situ irrigation water use measurements are hardly available. Although USGS provides

state-level data (e.g., 2000, 2005, 2010, 2015) and USDA offers county-level survey data (e.g., 2003, 2008, 2013, 2018), only

the 2018 dataset overlaps with the SMAP observation period. Moreover, the broader spatial aggregation and lower temporal

resolution of these datasets limit their applicability for direct comparison.

To address these limitations, we selected the estimated irrigation water use map provided by Zhang and Long (2021), hereafter



referred to as the ZL21 map. Derived from model simulations that integrate remote sensing-based evapotranspiration with

simulated root zone soil moisture, the ZL21 map offers high-resolution (1 km) monthly irrigation water use estimates for the

CONUS over the period 2000–2020. For the comparison, the ZL21 map was resampled to a 9 km EASE-Grid 2.0 and the

monthly data were aggregated to obtain the total irrigation water use during the cropping season. This dataset permits a more

direct and accurate large-scale comparison with our proposed irrigation signal map than the coarser state-level and county-

level data.

**2.2 Study area**

According to the validation reports for the SMAP Level 2/Level 3 product (O'Neill et al., 2020) and the SMAP L4 product

(Reichle et al., 2023a), these products are supported by the most extensive network of validation sites in CONUS, where their

accuracy is considered to be greatest. Given these considerations, the present study selected the Central Valley in California,

one of the largest irrigation regions in the world, as the primary study area. Additionally, experiments were conducted in Snake

River Plain and Nebraska High Plains to evaluate the broader applicability of the proposed method. Figures 1a and 1b illustrate

the geographical location and the map of irrigated areas of Central Valley and CONUS.

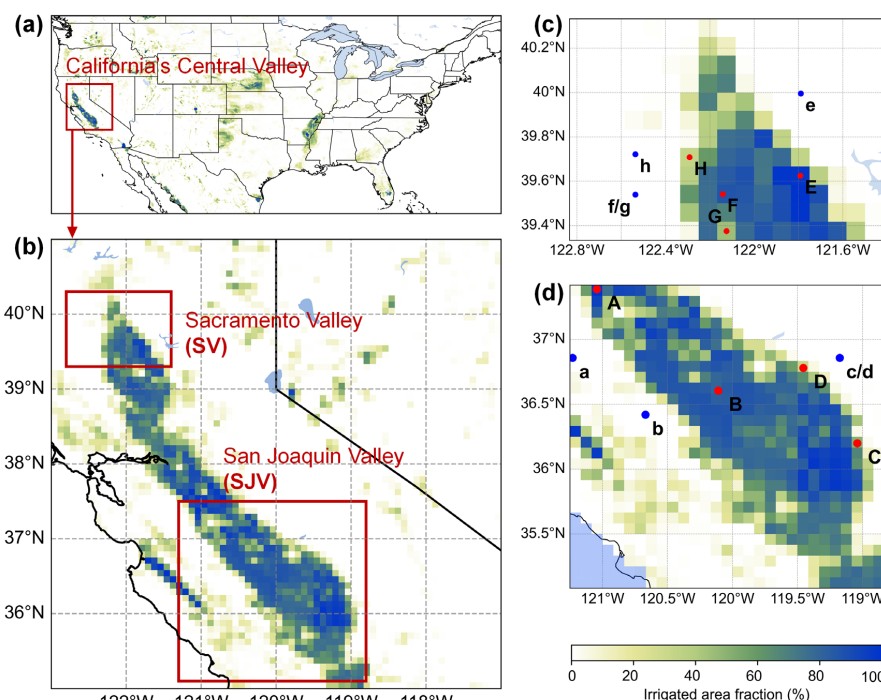

**Figure 1: Geographical locations of (a) CONUS, (b) California's Central Valley, (c) Sacramento Valley, and (d) San Joaquin Valley.**

**Shading illustrates irrigated area fraction from GMIA. Blue circles and lower-case letters in (c) and (d) denote non-irrigated grid**

**cells; red circles and upper-case letters represent irrigated grid cells; and light blue areas in (d) denote grid cells located over the**

**ocean.**

The United States National Agricultural Statistics Service identifies the Central Valley as the most irrigation-intensive region in CONUS, accounting for approximately 16% of the national total irrigation water usage (U.S. Department of Agriculture, 2024). Moreover, the Central Valley features different climate zones (Yang et al., 2017) and irrigation technologies, making it an ideal location to assess the performance of the proposed method under different conditions.

This study further divided the Central Valley into two subregions based on differing climate and irrigation practices: the southern San Joaquin Valley (SJV), which features a semi-arid climate and is predominantly equipped with sprinkler systems, and the northern Sacramento Valley (SV), characterized by a Mediterranean climate (Kottek et al., 2006) and primarily featuring flood irrigation systems. The main crops in SJV include almonds, grapes, and cotton, whereas rice dominates SV's farmlands. In fact, SV accounts for 95% of California's rice production and approximately 20% of the total rice production in CONUS (Singh et al., 2017). Given the high water demand of rice cultivation, SV relies more heavily on irrigation. In this region, rice farmlands are saturated from late April or early May, with planting commencing shortly thereafter, and they remain saturated with irrigation water until early September, when the farmlands are drained for harvest (Lawston et al., 2017).

## 3 Methodology

Identifying irrigation signals from SMAP L3_E and SMAP L4 is based on the assumption that the soil moisture difference between the two products stems from human irrigation activities under the following conditions: (1) the signal must be observed during the cropping season, (2) the bias between SMAP L3_E and L4 remains consistent across the cropping and non-cropping seasons, and (3) SMAP L3_E is significantly higher than L4. Note that these conditions allow for the presence of bias between SMAP L3_E and L4, as long as the bias remains consistent across the cropping and non-cropping seasons, which can be assessed in nearby grid cells that have no irrigation.

### 3.1 Evaluating the consistency between SMAP L3_E and L4

This section outlines how we evaluate the consistency between the SMAP L3_E and L4 products to ensure that the irrigation signal is genuine and not merely an artifact of systematic error in the L3_E and/or L4 product. This was accomplished by applying two criteria (described in the following paragraphs) to non-irrigated grid cells. The independent irrigated area map (GMIA) was used to locate the non-irrigated grid cells (defined as those with an irrigated area fraction below 0.1%). To reduce the sample size and improve clarity, a subsampling procedure was performed on these non-irrigated grid cells, involving two steps: (1) selection of all non-irrigated land grid cells as initially defined, and (2) uniform subsampling of these grid cells with a fixed interval. It is noted that this subsampling step is performed solely for clarity of presentation and does not influence the integrity or the overall analysis. Additionally, the crop calendar developed by Sacks et al. (2010) was employed to delineate



the cropping and non-cropping seasons.

The two criteria used to evaluate the consistency of two soil moisture products during the cropping and non-cropping seasons are as follows: The first criterion is the Pearson correlation coefficient $R$ across the entire period between the SMAP L3_E and L4 time series, which assesses the consistency in terms of the time series variability. A high $R$ value indicates that both products

exhibit similar variability in soil moisture dynamics and respond similarly to temporal variability in meteorological changes. We calculate $R$ across the entire time period and assume that the two products have consistent variability when $R > 0.70$. (Sensitivity analysis results for other thresholds, including 0.50, 0.60, and 0.80, are detailed in Supplementary Information Section S1.)

The second criterion is based on the Mean Difference ($MD$) between the SMAP L3_E and L4 data:

$$MD = \frac{1}{n}\sum_{i=1}^{n}(\theta_{L3,i} - \theta_{L4,i}) \tag{1}$$

where $n$ is the number of soil moisture samples (dimensionless) and $\theta_{L3,i}$ and $\theta_{L4,i}$ are the soil moisture values at time $i$ from SMAP L3_E and L4, respectively ($m^3/m^3$). We calculate $MD$ values for the cropping and non-cropping seasons individually in non-irrigated grid cells, and then compare the $MD$ values to assess whether the bias is consistent across both seasons. When the $MD$ values for the two seasons are within $\pm 0.04$ $m^3/m^3$, we assume that the mean soil moisture difference between the

two products is consistent across the two seasons and therefore largely unaffected by systematic error between the SMAP L3_E retrieval algorithm and the physical parameterizations of the L4 land surface model. The $\pm 0.04$ $m^3/m^3$ threshold for MD consistency used here equals the unbiased Root Mean Square Error ($ubRMSE$) accuracy requirement of SMAP soil moisture products. When the MD values are within $\pm 0.04$ $m^3/m^3$, the SMAP L3_E and L4 soil moisture are considered to be climatologically consistent across the cropping and non-cropping seasons in non-irrigated grid cells.

Next, we hypothesize that the soil moisture climatology is consistent among irrigated and non-irrigated grid cells within the study area. That is, if non-irrigated grid cells exhibit consistent seasonal bias across the cropping and non-cropping seasons, a similar consistency is assumed for irrigated grid cells.

**3.2 Identification of SMAP's irrigation signals**

After confirming the consistency of SMAP L3_E and L4 at non-irrigated grid cells, the irrigation signals can be identified as

the difference in the $MD$ between the cropping and non-cropping seasons. Quantitatively, we define the Irrigation Signal ($IS$), separately for each grid cell, as the difference between the $MD$ values during the cropping and non-cropping seasons:



$$IS = MD_{cro} - MD_{non} \tag{2}$$

where $MD_{cro}$ and $MD_{non}$ denote the L3_E-minus-L4 mean difference for the same grid cell during cropping and non-cropping

seasons, respectively ($m^3/m^3$). A larger $IS$ value indicates a higher irrigation intensity for the grid cell. Note that $IS$ represents

irrigation intensity rather than the absolute irrigation water use amount; this distinction is further elaborated in Section 5.

We applied the quantitative $IS$ method across the entire study area to map irrigation intensity, excluding the grid cells that do

not meet the climatological consistency criteria. Recall that for the consistency evaluation in non-irrigated grid cells, we

compute $R$ across the entire year. For the $IS$ map, however, we calculated $R$ only during the non-cropping season ($R_{non}$) because

irrigation during the cropping season is expected to elevate soil moisture in irrigated grid cells, which is captured in SMAP

L3_E but not in L4. To validate our results, the multi-year average $IS$ map is compared with both the GMIA map and the ZL21

map.

## 4 Results

### 4.1 Verification of SMAP L3_E and L4 data consistency

To verify climatological consistency between SMAP L3_E and L4 across the cropping and non-cropping seasons, soil moisture

estimates of non-irrigated grid cells from 2016 to 2022 were analyzed. Within the SJV and SV regions, a total of 32 grid cells

were subsampled from the pool of non-irrigated grid cells (irrigated area fraction below 0.1%) to ensure representative

coverage (Supplementary Information Section S2). For brevity, only six selected grid cells are shown in the main text. Figure

1c, 1d, and Table 1 display the locations of these non-irrigated grid cells (blue circles) along with their irrigated area fractions

as reported by the GMIA.

For further comparison in the next section, the corresponding irrigated grid cells (red circles) of each selected non-irrigated

sample are presented. These grid cells were selected by two criteria: (1) geographic proximity to the sampled non-irrigated

grid cells and (2) irrigation fractions that closely match target fractions—near 100%, 80%, 50%, and 30% as reported by the

GMIA. In cases where multiple irrigated grid cells were in very close proximity (e.g., grid cells C and D; or F and G), they

were paired with the same non-irrigated grid cell (e.g., grid cells c/d; or f/g). Additionally, as a sensitivity test, 10 alternative

pairs were analyzed (see Supplementary Information Section S3) to verify the robustness of our hypothesis.



**Table 1: Irrigated area fractions from GMIA for selected non-irrigated and irrigated grid cells**

| Region | Non-irrigated grid cells | | Irrigated grid cells | |
|---|---|---|---|---|
| | Grid cell No. | Irrigated area fraction | Grid cell No. | Irrigated area fraction |
| SJV | a | 0.00% | A | 99.70% |
| | b | 0.11% | B | 80.02% |
| | c/d | 0.00% | C | 50.79% |
| | | | D | 30.20% |
| SV | e | 0.00% | E | 92.11% |
| | f/g | 0.00% | F | 81.05% |
| | | | G | 42.14% |
| | h | 0.00% | H | 30.97% |

Figure 2 displays the soil moisture time series from SMAP L3_E and L4 for the six selected non-irrigated grid cells in 2016.

The cropping season (May to October) and the non-cropping season (November to April) are defined based on crop calendars from Sacks et al. (2010). Visual inspection indicates that the temporal variabilities of SMAP L3_E and L4 correspond reasonably well throughout the year. Specifically, both retrievals exhibit high variability during the non-cropping season when a considerable amount of precipitation occurs, while dropping to a low, quasi-constant level during the cropping season when precipitation is scarce.

A quantitative evaluation using $R$ and $MD$ suggests that SMAP L3_E and L4 are systematically consistent in the Central Valley. In all selected non-irrigated grid cells, $R$ exceeds 0.70 and the difference between the cropping season $MD$ ($MD_{cro}$) and the non-cropping season $MD$ ($MD_{non}$) remains within $\pm 0.04$ m³/m³. (Time series of differences between SMAP L3_E and L4 in non-irrigated grid cells are provided in the Supplementary Information Figure S4-1.) As detailed in the Supplementary Information Section S2, 93.8% (30 out of 32) sampled non-irrigated grid cells meet these criteria. Furthermore, a paired t-test

showed no statistically significant difference between $MD_{non}$ and $MD_{cro}$ ($p$-value > 0.05), indicating that SMAP L3_E and L4 exhibit similar responses to the temporal variability and maintain a consistent bias across the cropping and non-cropping seasons.



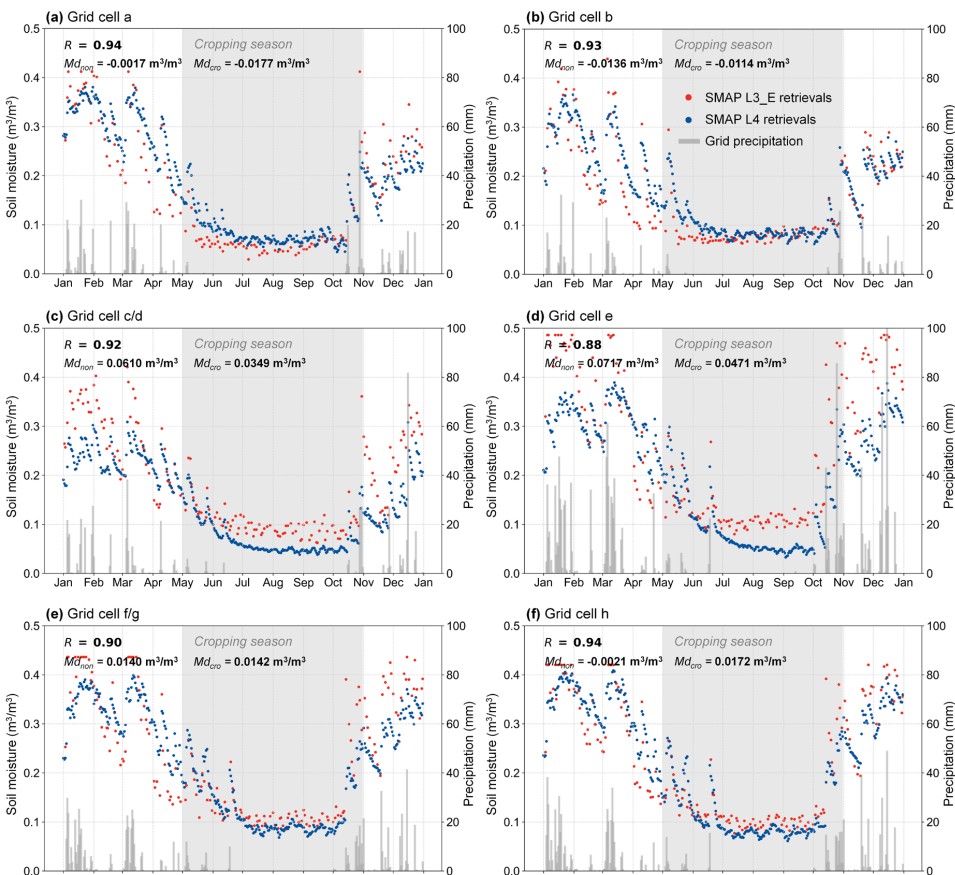

**Figure 2: SMAP surface soil moisture from (red dots) L3_E and (blue dots) L4 for selected non-irrigated grid cells in 2016 (left axis).**
**Gray bars indicate precipitation (right axis), and light gray shaded area represents the cropping season. See Figure 1 for grid cell locations.**

**4.2 Verification of detected irrigation signals at the point scale**

**4.2.1 San Joaquin Valley**

Figure 3 presents the SMAP L3_E and L4 soil moisture for four irrigated grid cells in SJV in 2016. The temporal variability of the two products is climatologically consistent during the non-cropping season, with Pearson correlation coefficient $R_{non}$ exceeding 0.70. Supplementary Information Figure S5-3 reinforces this observation, which shows that 100% (18 out of 18) uniformly sampled irrigated grid cells have $R_{non}$ values above 0.70, confirming the consistency of both products. At the same time, both visual inspection and statistical analysis indicate that, while the two soil moisture time series are similar during the

non-cropping season, a significant gap emerges during the cropping season. Specifically, $MD_{cro}$ is much higher than $MD_{non}$, a difference confirmed as statistically significant by a paired t-test (detailed in the following paragraph). In the absence of





precipitation during the cropping season, the higher soil moisture level in SMAP L3_E suggests that the additional soil water

is likely owing to human irrigation activities.

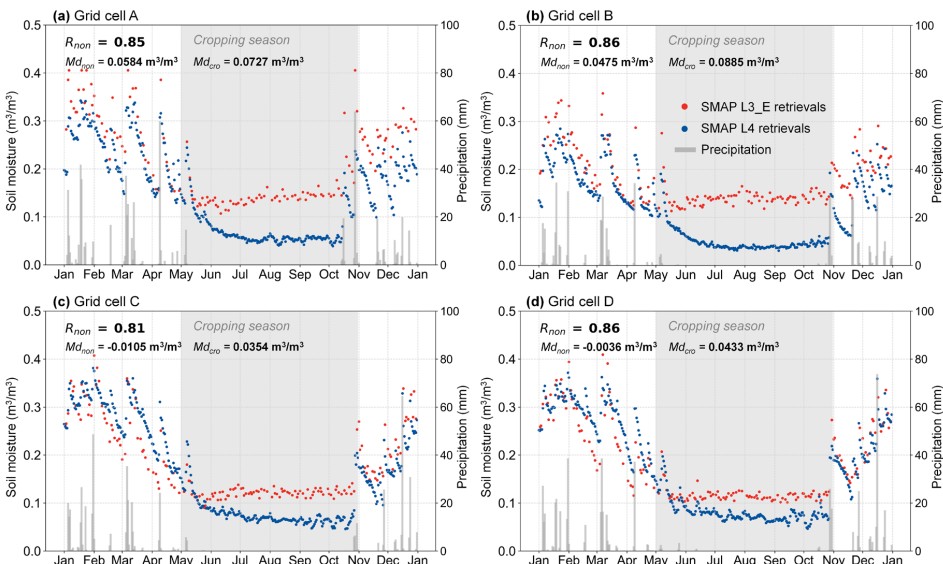

**Figure 3: As in Figure 2 but for selected irrigated grid cells in SJV, 2016.**

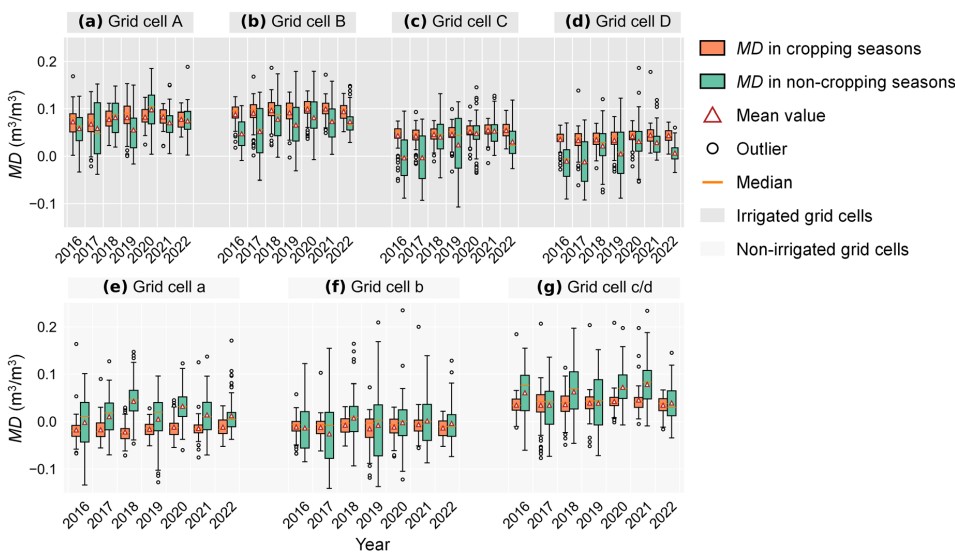

**Figure 4: Distribution of *MD* values in the (red) cropping and (green) non-cropping seasons for selected (top row) irrigated and**

**(bottom row) non-irrigated grid cells in SJV from 2016 to 2022. See Figure 1 for grid cell locations.**



To quantitatively demonstrate the presence of irrigation signals at these grid cells, we compared $MD$ of the SMAP L3_E and L4 between the cropping and non-cropping seasons in SJV from 2016 to 2022 (Fig. 4). A paired $t$-test was conducted to assess whether the $MD$ values differ significantly between the two seasons (time series of differences between SMAP L3_E and L4 in selected irrigated grid cells are provided in the Supplementary Information Figure S4-2). At the irrigated grid cells (Figs. 4a–d), we generally observed higher $MD$ values during the cropping season compared to those during non-cropping season ($p$-

value < 0.05), although exceptions exist, such as grid cell A in 2020. In contrast, the cropping and non-cropping gaps of $MD$ at the non-irrigated grid cells (Figs. 4e–g) are close enough to be neglected: the $MD$ difference between the two seasons are generally within $\pm 0.04$ m³/m³. The paired $t$-test results ($p$-value = 0.2056 > 0.05) reinforce that there is no statistically significant difference of the $MD$ values between the two seasons. Together, these results quantitatively demonstrate that our identification method is valid, indicating that the significant soil moisture gaps during the cropping season at irrigated grid

cells are likely due to irrigation rather than other systematic differences between the two SMAP products.

### 4.2.2 Sacramento Valley

Figure 5 presents the SMAP L3_E and L4 soil moisture for four irrigated grid cells in SV in 2016. In contrast to the results for SJV, we observed that not all regions in the Central Valley exhibit high climatological consistency between SMAP L3_E and L4. For example, the three selected irrigated grid cells in SV (Figs. 5a-c and 6a–c) show inconsistent variability during the

non-cropping season ($R_{non}$ < 0.70). In Supplementary Information Figure S5-3, we show that 72.7% (8 out of 11) uniformly sampled irrigated grid cells in SV have $R_{non}$ below this threshold. This decrease in $R_{non}$ is attributed to the SMAP L3_E algorithm producing several saturated soil moisture points (Figs. 6a–c, Supplementary Information Figure S2-4) during the non-cropping season, while SMAP L4, although also indicating high soil moisture levels, remains unsaturated. Note that these grid cells will be classified as having inconsistent climatology in the subsequent irrigation signal map analysis.



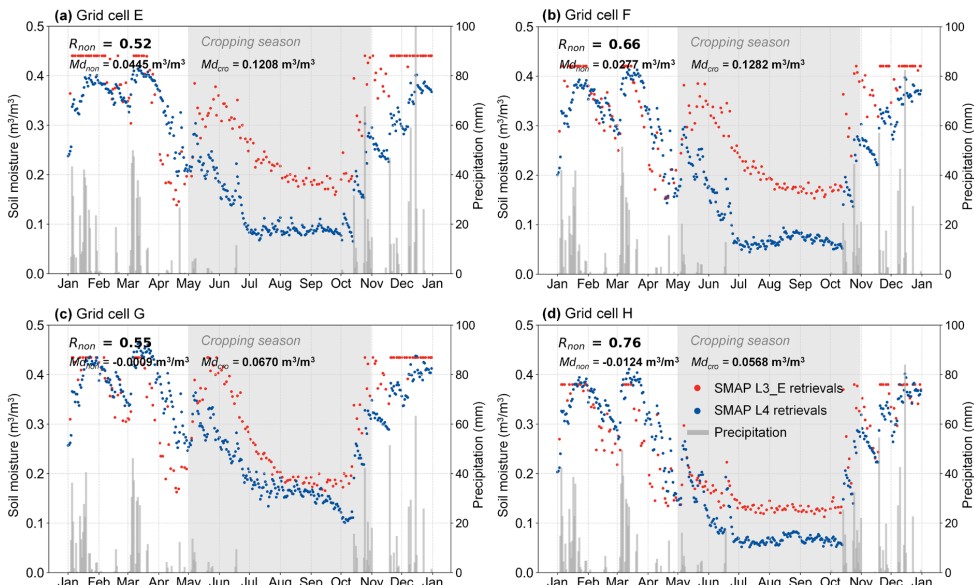

**Figure 5: As in Figure 2 but for selected irrigated grid cells in SV, 2016.**

Furthermore, Figure 6 compares *MD* between the cropping and non-cropping seasons. A marked increase in *MD* during the cropping season was observed in irrigated grid cells (Figs. 6a–d), with SMAP L3_E soil moisture retrieval showing a more significant increase during the cropping season. This is likely driven by the high irrigation water demand associated with large-scale rice cultivation in SV (Wong et al., 2021).

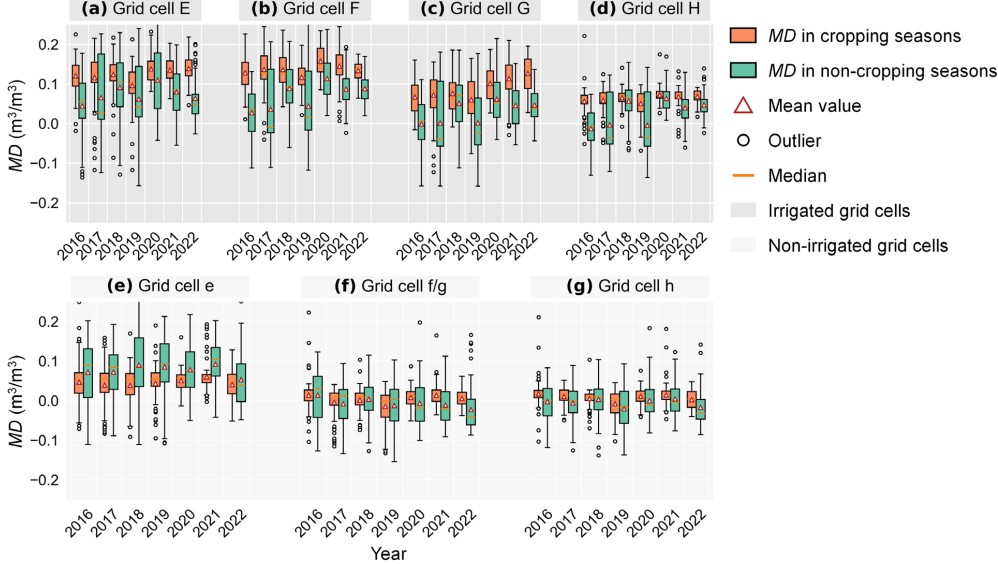

**Figure 6: As in Figure 5 but for selected grid cells in SV.**





### 4.3 Map of SMAP's irrigation signals in San Joaquin Valley

Having confirmed the climatological consistency between the two satellite products and verified the validity of the irrigation

detection method at several point locations, we now present a map of SMAP-based irrigation intensity. Figure 7 shows the

estimated average *IS* for SJV from 2016 to 2020 (Fig. 7a), alongside the corresponding irrigated area fraction from the GMIA

(Fig. 7b) and the average irrigation water use derived from the ZL21 map (Fig. 7c). To maintain conciseness, the estimated *IS*

map for SV is provided in Supplementary Information Section S6. The elliptical irrigation signal shape, extending from

northwest to southeast, can be identified in SJV. In addition, localized irrigation signals are detected in some smaller farmlands,

such as the regions (i) Salinas Valley and (ii) most southern part of SJV in Fig. 7, which are also recognized as irrigated in the

GMIA and the ZL21 map. In the northeast of SJV, at the edge of Sierra Nevada mountains (Supplementary Information Figure

S7-1), we observe a small fraction of grid cells where the non-cropping season soil moisture time series is not consistent

(shown in gray shading in Fig. 7a). This inconsistency is attributed to the high elevation in this region, where soil remains

frozen or snow-covered during the non-cropping season (primarily winter and spring). Under such conditions, the microwave

radiative transfer model used in the SMAP L3_E retrieval algorithm and the L4 model assimilation cannot properly simulate

the link between soil moisture and brightness temperature. Consequently, the L3_E algorithm flags the retrievals for

questionable quality, and the L4 algorithm does not assimilate the SMAP-observed brightness temperature.



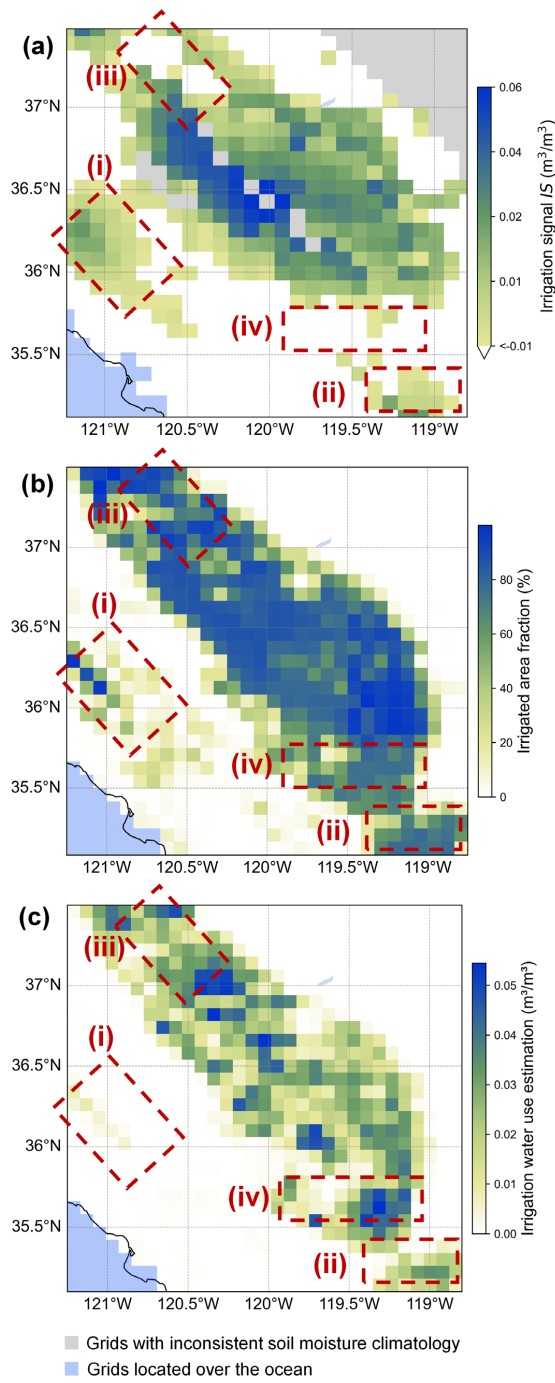

**Figure 7: Map of irrigation signals (*IS*) and the validations in SJV. (a) Average *IS* estimated from SMAP for 2016–2020, (b) irrigated**

**area fraction from the GMIA, and (c) average irrigation water use estimations for 2016–2020 from the ZL21 map. Light blue areas**

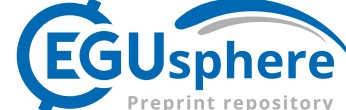

denote grid cells located over the ocean, while gray colored areas indicate grid cells with inconsistent soil moisture climatology. Note: While both the *IS* map and the ZL21 map (e.g., Figs. 7a, 7c) share the same units, they differ in their interpretation. The ZL21 map quantifies irrigation water use per cubic meter, whereas the *IS* map primarily captures the accumulated soil moisture differences

between SMAP L3_E and L4 products, making their absolute magnitudes non-comparable and relevant only in terms of intensity.

Figure 8 displays the scatterplot and *R* values between the estimated average *IS* map and the irrigated area fraction from the GMIA (Fig. 8a) as well as the average irrigation water use estimations from the ZL21 map (Fig. 8b). Sensitivity analysis scatterplots for different soil moisture variability consistency thresholds are detailed in Supplementary Information Section S8.

The scatterplot for two benchmark products irrigated area fraction and average irrigation water use estimations is shown in Fig. 8c. The spatial correlation with the GMIA and ZL21 map are 0.66 and 0.50, respectively. These results further suggest that our proposed method–where only SMAP L3_E and L4 soil moisture are used–is valid and SMAP products do have the potential to reveal irrigation effects.

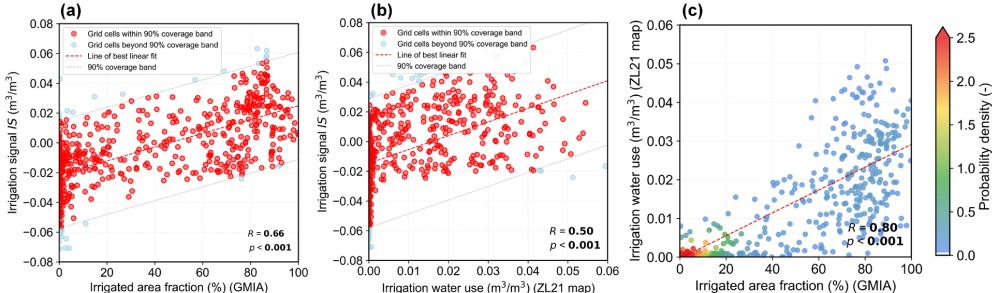

**Figure 8: Scatterplot of grid cell values in the *IS* map compared with those from the GMIA and the ZL21 map in SJV. (a) Comparison between the *IS* and irrigated area fraction (GMIA), (b) comparison between the *IS* and the irrigation water use (ZL21 map), and (c) comparison between the two benchmark products: the irrigated water use (GMIA) and the irrigation water use (ZL21 map).**

However, some discrepancies remain between the irrigated area (GMIA), the irrigation water use estimation (ZL21 map), and

the proposed *IS* map. First, the spatial gradients in the *IS* map derived from SMAP tend to be smoother, likely due to its effective spatial resolution being coarser than the nominal 9 km EASE-Grid, as the native radiometer footprint (~36 km) is interpolated to the nominal resolution by the Backus-Gilbert optimal interpolation algorithm (ONeill et al., 2023). This smoothing could explain why some grid cells along the boundaries of irrigation regions show near-zero values in the GMIA and ZL21 maps but non-zero values in the *IS* map in Fig. 8. Second, the satellite-based results strongly reflect the local

characteristics. For instance, over grid cells in region (iii) (around 37.3°N, 120.7°W), the GMIA indicates high irrigated

fractions, whereas both the ZL21 map and the proposed *IS* map show weak irrigation effects. This discrepancy is probably caused by intensive urbanization in this area, as directly observed on Google Earth (Supplementary Information Section S9); since the GMIA was estimated based on the global irrigated land data in 2005 (Siebert et al., 2013), any subsequent changes are not captured in the GMIA. Nonetheless, limitations also exist in the satellite-based approach. In region (iv) (around 35.6°N, 119.5°W), the *IS* map shows little or no irrigation signal, despite both the GMIA and the ZL21 map indicating heavy irrigation. This discrepancy could be due to the prevalence of developed subsurface irrigation systems in this area (Faunt, 2009; Sorooshian et al., 2014). The estimated *IS* map relies solely on satellite-based surface soil moisture so that it cannot monitor the water that transfers directly to the root zone soil.

**5 Discussion**

This study demonstrates the feasibility of detecting irrigation effects from SMAP soil moisture products in the Central Valley. Applying the approach to monitor irrigation water use on a global scale still faces several noteworthy technical challenges. These challenges include ensuring the effectiveness of detection methods across diverse climatic conditions and addressing the limitation of our method, which can identify whether a region is irrigated and estimate relative intensity, but not quantify the exact amount of irrigation water used. In this section, we discuss these issues based on our extended analyses and a review of relevant literature, aiming to provide insights for future large-scale irrigation estimation studies based mainly on satellite-derived soil moisture.

**5.1 Evaluation of detection effectiveness across different climatic conditions**

Our results indicate that the proposed method based on mean differences between SMAP L3_E and L4 data works reasonably well in revealing irrigation signals in the Central Valley, where dry summer and marginal precipitation during the cropping season make these signals distinct. In such regions with dry summer climates (e.g., (semi-)arid or (semi-)Mediterranean climates), increases in soil moisture containing irrigation effects can be confidently attributed to irrigation activities. However, other regions may experience year-round precipitation or precipitation overlapping with the cropping season. To further evaluate the effectiveness of our method under these conditions, we extended our analysis to two additional irrigation regions in CONUS: the Snake River Plain (SRP) in Idaho, and the Nebraska High Plains (NHP). SRP is characterized by a cold semi-arid climate (Kottek et al., 2006) with marginal precipitation during both the cropping season and summer, and Idaho is the second-largest state in CONUS in terms of irrigation water withdrawals (Murray, 2018). In contrast, NHP lies in a transitional climatic zone: its western part is semi-arid, while its eastern part features a humid continental climate with year-round precipitation (Kottek et al., 2006). The primary crops in NHP are corn and soybean, with irrigation requirements for corn decreasing from approximately 350 mm per year in the western NHP to about 150 mm per year in the eastern NHP (Zaussinger





et al., 2019).

Based on our method, we expect the *MD* between two SMAP products to be higher during the cropping season than during the non-cropping season for the irrigated grid cells. In SRP, soil moisture behavior aligns with this expectation, with *MD* significantly higher from June to October (Fig. 9). However, in NHP, the *MD* difference is only marginal due to consistent precipitation throughout the year (Fig. 10). The proposed *IS* maps for SRP and NHP are provided in Supplementary Information

Section S10. These results indicate that our method can only effectively capture irrigation signals in regions with limited cropping season precipitation (e.g., (semi-)arid, (semi-)Mediterranean climatic regions). The challenge of precipitation interfering with the detection of irrigation effects in regions where precipitation overlaps with irrigation is not new in satellite-based irrigation detection studies. For example, Lawston et al. (2017) reported clear irrigation effects from SMAP L3 products in Sacramento Valley and San Luis Valley, where (semi-)Mediterranean climates with dry summers and autumns coincide with

cropping seasons. However, eastern Nebraska, the same region as we tested in Figure 10, exhibited blurred irrigation effects due to overlapping precipitation. Similarly, Zaussinger et al. (2019) found that their SMAP-based irrigation estimations are weak in precipitation-dominated irrigation regions like the Lower Mississippi Floodplain and NHP.

One potential solution is to explicitly account for the residual influence after precipitation. For example, estimating how much water from precipitation can be retained in the surface soil layer (e.g., applying the precipitation fraction concept as proposed

by McColl et al. (2017)) and subtracting it from the soil moisture observations. Integrating evapotranspiration is another promising approach, although it remains challenging due to the limited availability of reliable large-scale observations.

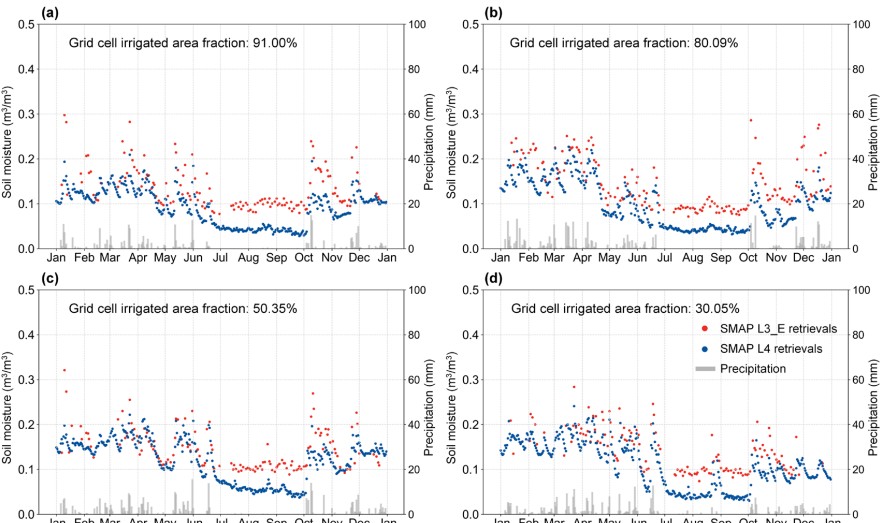

**Figure 9: As in Figure 2 but for irrigated grid cells in SRP, 2018.**



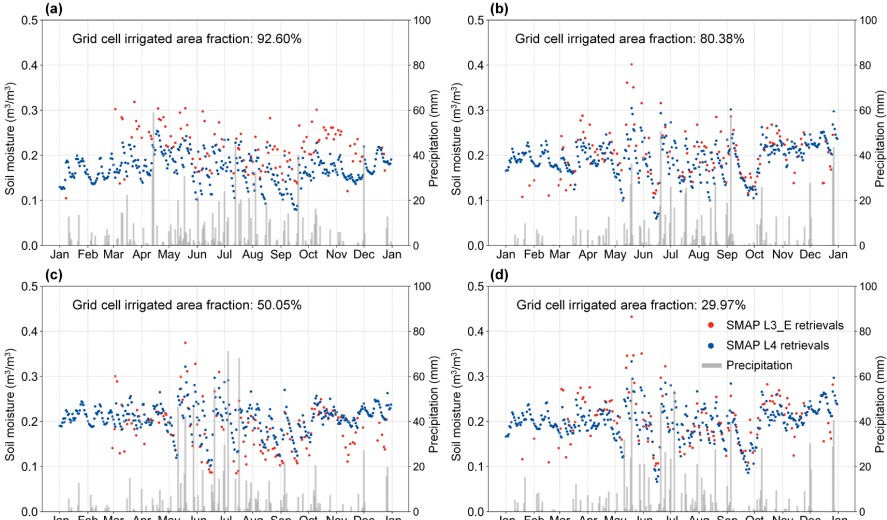

**Figure 10: As in Figure 2 but for irrigated grid cells in NHP, 2018.**

Grid cells with inconsistent climatology between SMAP L3_E and L4 were masked and excluded from the analysis. In this study, these inconsistencies primarily arise in high-elevation mountainous areas characterized by snow cover, where the temporal variability between the two datasets diverges. Including these grid cells could potentially lead to erroneous

interpretations when detecting irrigation signals. Future studies could incorporate additional geographic information, such as elevation and land-use classifications, to better distinguish between irrigated grid cells and high-elevation non-irrigated grid cells, thereby enhancing the reliability of irrigation detection.

### 5.2 Algorithm complexity for absolute irrigation water amount calculation

Another limitation of this study is that we provide irrigation intensity rather than absolute irrigation water use amounts. This

is because satellite-based soil moisture retrievals, such as those from SMAP, represent a stock variable that reflects the integrated effect of various flow processes such as precipitation, evapotranspiration, runoff, and irrigation. In essence, estimating individual components (e.g., irrigation water, runoff, evapotranspiration) of the total flux directly from knowledge of the stock variable is mathematically infeasible. Additionally, factors such as noise, the representativeness of soil depths in satellite retrievals, bias in reference soil moisture data, and the inherent soil moisture memory further complicate the estimation

of absolute irrigation water use amounts (Brocca et al., 2018; Foster et al., 2020; Jalilvand et al., 2019, 2023).

To address these challenges, one promising approach is to integrate physical and statistical methods to better characterize the nonlinear soil moisture processes. Specifically, two key water sources should be separated and modeled: (1) precipitation- and

irrigation-caused soil moisture increments and (2) noise and bias versus real soil moisture signals. On the other hand, data-driven machine learning methods offer a promising alternative to rapidly "learn" intricate relationships between the "true" irrigation water amount and soil moisture (noting that, due to the scarcity of observed data, the "true" amount referenced here is typically estimated from validated simulation models). This approach has been applied in regional studies as seen in China (Liu et al., 2024), the state of Kansas (Wei et al., 2022), and other regions. With similar applications extended to other regions, such maps would provide valuable insights for hydrological modeling and support sustainable water resource management on a global scale.

## 6 Conclusions

In this study, we proposed a method to detect irrigation signals directly from SMAP L3_E and L4 soil moisture products. A critical foundation of this approach is that SMAP L4 inherently excludes irrigation effects, a key finding confirmed by our systematic comparison with SMAP Level 3 products. This exclusion arises because SMAP L4 assimilates only brightness temperature anomalies from SMAP Level 1 observations, where irrigation-induced changes are absorbed into the climatological seasonal mean rather than being treated as anomalies. Our analysis confirms that SMAP L4 soil moisture does not include irrigation effects, making it a reliable baseline for isolating anthropogenic water use in water resource studies.

By analyzing mean differences during the cropping and non-cropping seasons in Central Valley California, we demonstrate that SMAP can effectively identify irrigation effects in regions with (semi-)Mediterranean and (semi-)arid climate, where precipitation during the cropping season is limited. Our application to the Snake River Basin further confirms that the method is effective for regions with similar climatic conditions. However, in regions such as the Nebraska High Plains and the Lower Mississippi Floodplain, characterized by year-round precipitation or overlapping precipitation and cropping season, the method struggles to separate irrigation effects from precipitation, rendering it ineffective. We also provide a spatial irrigation signal map that indicates the irrigation water use intensity. The spatial correlations between our maps and two benchmark maps demonstrate reasonable agreement, further underscoring the validity of SMAP-based soil moisture retrievals for irrigation monitoring. Compared to other satellite-based irrigation studies that used or developed comprehensive models, our method is built solely on statistical analyses of two satellite-based pre-tuning soil moisture datasets and shows high simplicity with minimum requirement of additional data.

One noteworthy limitation of our method is that we cannot explicitly quantify the absolute irrigation water use amount, which hinders its direct application in agricultural water resource management and the validation for hydrological simulations. This limitation stems from the simplicity of our mathematical approach, which makes it challenging to derive flow variables (i.e., irrigation water) from stock measurements (i.e., soil moisture).

Nonetheless, the fact that SMAP has the potential to reveal irrigation event occurrences provides a starting point for various applications. For example, the occurrence of irrigation activities can vary annually depending on the local agricultural management or land use changes. Such information cannot be captured by the static irrigated area map but could be revealed

by SMAP. One promising direction is the development of quasi-real-time irrigation occurrence maps based on dynamic SMAP estimations to improve the realism of current hydrological monitoring. Another direction is to incorporate machine learning tools to establish the relationship between monthly SMAP irrigation signals and limited observed irrigation water use amount data. By extending the relationship, it is promising to obtain water use amount data in observation-scarce regions.

**Data availability**

SMAP L3_E and L4 products are freely accessible via NASA Earthdata portal (http://earthdata.nasa.gov/). The Global Map of Irrigation Areas can be obtained through the FAO website (Geospatial Information). Irrigation water use estimates are available from https://doi.org/10.1029/2021WR031382 (Zhang and Long., 2021).

**Author contributions**

H. X. and Q. H. designed the study and wrote the original draft. R. R. proposed the research idea and contributed to manuscript

revision. N. H. and T. O. critically reviewed and edited the manuscript. All authors discussed experiments and results, provided substantial feedback, and approved the final version of the manuscript.

**Competing interests**

The authors declare that they have no conflict of interest.

**Acknowledgements**

This work was partially supported by Japan Society for the Promotion of Science (KAKENHI; 21H05002) and the Environment Research and Technology Development Fund (JPMEERF23S21120) of the Environmental Restoration and Conservation Agency, Ministry of the Environment of Japan. Q. H. appreciates financial support from Japan Society for the Promotion of Science International Research Fellow Program. R. R. was supported by the NASA SMAP Project. The authors thank Dr. Stefan Siebert for making the Global Map of Irrigated Area openly available, and Dr. Caijin Zhang and Dr. Di Long



for providing access to the CONUS irrigation water use data. The authors also acknowledge the invaluable assistance from the

NASA SMAP science team.

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
