# Peer review of "Detecting irrigation signals from SMAP L3 and L4 soil moisture: A case study in California's Central Valley"

_EGUsphere, 2025_

## Referee Comment (RC2)

**Detecting irrigation signals from SMAP L3 and L4 soil moisture: A case study in California's Central Valley**

The authors propose an approach for detecting irrigation signals through discrepancies between SMAP L3 (potentially able to track irrigation) and L4 data (unable to track irrigation). The strength of the methodology consists in solely relying on data without the need of any modeling. Analyses are performed rigorously and results are clearly presented. Nevertheless, the study appears as a bit out-of-date with respect to the current status of the art in terms of irrigation monitoring through satellite data. In fact, in light of several well-established methodologies for detecting in space and time irrigation events and for estimating irrigation water use, with some of them even close to facing operational implementation, the current study appears limited in its scope. In addition, the capability of SMAP retrievals in detecting irrigation in California has been already proved in previous studies (see, e.g., https://doi.org/10.1002/2017GL075733, https://doi.org/10.1016/j.hydroa.2023.100169). On top of this, in addition to limitations discussed by the authors, those linked to the mismatch between the spatial resolution of SMAP retrievals and the extent of irrigated areas elsewhere are not mentioned but represent a critical point in the irrigation detection domain (https://doi.org/10.1016/j.jag.2022.102979). In my opinion, the limited scope of this paper with respect to the current status of knowledge does not incentivize its publication. The paper does not propose an irrigation quantification method (because of the limits in retrieving irrigation fluxes clearly explained by the authors) neither an irrigation mapping approach (as the a priori knowledge of irrigated and non-irrigated pixels is required). It could be seen as a method for detecting irrigation events but definitely an effort is required for highlighting advantages with respect to previous studies (e.g., https://doi.org/10.3390/rs15051449 or https://doi.org/10.3390/rs12091456, to cite a few). Please find some specific comments as follows:

L 21: To what temporal resolution do the correlation coefficients refer?

L 62-75: SM-based methodologies for retrieving irrigation information can be divided into two main categories, namely baseline approaches (as for instance https://doi.org/10.5194/hess-23-897-2019, https://doi.org/10.3390/rs13091727) or methodologies based on the soil water balance (e.g.,

https://doi.org/10.5194/essd-15-1555-2023). Note that such methodologies led to the development of satellite-based irrigation water use datasets (https://doi.org/10.5281/zenodo.8086046), also available for the US (https://doi.org/10.5281/zenodo.14988198).

L 90-92: So why California only is mentioned in the title?

L 170: performances of ZL21 should be reported to understand its reliability as a comparative dataset.

L 195: is flood irrigation an issue for detecting the irrigation signal?

Figure 2: grid cell e) seems to show slightly different dynamics.

L 309: what is the entity of MD discrepancies?

Figure 7: panel c), how il this map converted to m3/m3? Is porosity taken into account?

L 420-421: what about GLEAM, Sen-ET, …

---

## Author Comment (AC1)

**Reply on Comment (Community Reviewer #1)**

We sincerely thank the community reviewer for volunteering the time to read our manuscript and offer constructive suggestions.

Below, each comment from the community reviewer is *in italicized font*, followed by our responses in red normal font. Any new or added text in the manuscript is underlined in red, deleted text is with  in red, and these changes will be incorporated into the next revision.

All the line numbers in this reply refer to the original version of EGUsphere Manuscript ID: **egusphere-2025-2004**

**Comments #1:**

*The manuscript presents a novel and promising approach for detecting irrigation signals using SMAP (Soil Moisture Active and Passive) Level 3 Enhanced (L3_E) and Level 4 (L4) soil moisture products, with a focus on the Central Valley, California, and extended analyses in the Snake River Plain and Nebraska High Plains. The method leverages the difference between satellite-observed (L3_E) and model-assimilated (L4) soil moisture data to isolate irrigation effects, capitalizing on the fact that SMAP L4 excludes irrigation signals due to its assimilation of brightness temperature anomalies. The study is well-structured, clearly written, and makes a compelling case for its simplicity and minimal reliance on additional data or complex model tuning compared to previous approaches.*

**Reply:** We thank the community reviewer for your comments and for recognizing the novelty and promise of our approach. We have carefully revised the manuscript based on your comments and those from the two anonymous referees.

**Comments #2:**

*The introduction could better contextualize the novelty of the proposed method by*

*briefly summarizing how it differs from prior soil moisture-based studies (e.g., Zaussinger et al., 2019; Lawston et al., 2017) beyond the mention of avoiding complex model tuning. A concise statement on how the use of SMAP L3_E and L4 together is unique would strengthen the rationale. Add a sentence or two explicitly stating how the proposed method advances beyond existing soil moisture-based approaches, particularly in terms of leveraging SMAP's internal products to ensure climatological consistency.*

**Reply:** Thanks for your valuable comments. We agree with your suggestions that the Introduction should more clearly contextualize the novelty of proposed method.

We will add some descriptions in the next version, and the revised parts will be added as follows:

Line 89: Compared to the prior soil moisture-based studies, the key innovation of this study lies in its ability to maximize the preservation of climatological consistency of existing products without post-processing. The proposed approach minimizes the risk of introducing systematic biases, allowing the detected signals to be reliably attributed to irrigation effects rather than other factors.

**Comments #3:**

The explanation of the SMAP L4 assimilation process is technically dense and may be challenging for readers unfamiliar with data assimilation. A simplified summary could improve accessibility.

**Reply:** Thanks for your suggestions. We understand it could be challenging for readers who are not familiar with data assimilation and SMAP mission.

To make the explanation easier and clearer, we will add a simplified summary followed by the original technical description. The revised parts will be added as follows:

Line 148: In short, SMAP L4 does not directly assimilate absolute brightness

temperatures Tb from the SMAP Level 1 product; it assimilates only their anomalies relative to a mean seasonal cycle, and because irrigation activities in large agricultural regions are often similar from year to year, their signal is absorbed into the mean seasonal cycle rather than appearing as an anomaly.

**Comments #4:**

*The choice of the ±0.04 m³/m³ threshold for MD consistency is based on the unbiased RMSE accuracy requirement of SMAP products, but its suitability for irrigation detection could be further justified, as irrigation signals may vary in magnitude across regions.*

**Reply:** Thank you for the comment. We adopted a ±0.04 m³/m³ threshold to preliminarily examine whether the mean difference (*MD*) between cropping and non-cropping seasons is consistent or not. (Recall that *MD* is defined as the difference between the SMAP L3 and L4 products, and that our aim is to understand if this interproduct difference remains consistent across the cropping and non-cropping seasons.)

Crucially, we apply a paired t-test at the 5% significance level to assess *MD* consistency across the cropping and non-cropping seasons. For each grid cell, the null hypothesis (H₀: $MD_{cropping} = MD_{non-cropping}$) is tested. If the test yields $p > 0.05$, we do not reject H₀ and thus regard the *MD*s as statistically consistent between the cropping and non-cropping seasons; if $p \leq 0.05$, we reject H₀ and conclude that the *MD* values differ significantly between the two. We will revise the following descriptions in the next version:

Line 229: A paired *t*-test is applied across years to test the statistical differences of *MD* values between the cropping and non-cropping seasons. When accepting the null hypothesis and the *MD* values difference for the two seasons are within ±0.04 m³/m³, we assume that the mean soil moisture difference between the two products is consistent

across the two seasons and therefore largely unaffected by systematic error between the SMAP L3_E retrieval algorithm and the physical parameterizations of the L4 land surface model.

Line 233: When accepting the null hypothesis and differences  are within $\pm 0.04$ m$^3$/m$^3$, the SMAP L3_E and L4 soil moisture are considered to be climatologically consistent across the cropping and non-cropping seasons in non-irrigated grid cells.

**Comments #5:**

*The discussion of discrepancies between the IS map, GMIA, and ZL21 map is thorough but could better address why the IS map fails to detect irrigation in areas with subsurface irrigation (e.g., region iv). A brief explanation of how SMAP's 5 cm penetration depth limits detection of deeper irrigation could clarify this.*

**Reply:** Thank you for your valuable suggestions. SMAP's 5 cm penetration depth indeed limits the detection of deeper irrigation, which is a challenge that is common to all irrigation detection or estimation studies that are based on soil moisture retrievals from space-borne observations.

We will add the following explanation in the next version:

Line 383: Specifically, the nominal SMAP detection depth is top 5 cm of the soil column, so the irrigation water delivered below this depth, directly into the root zone, may not produce an immediate surface signal, causing the *IS* map to under-represent irrigation where water bypasses the near-surface layer.

**Comments #6:**

*The conclusions could better highlight the study's contribution to the field, such as its advancement over previous satellite-based methods or its potential to improve hydrological modeling. I strongly recommend authors to cite papers such as:*

*Assimilation of Sentinel-Based Leaf Area Index for Modeling Surface-Ground Water Interactions in Irrigation District*

**Reply:** Thank you for your suggestions.

We will carefully review this paper and find its potential complements to our approach, and we will cite this paper in the next version.

---

## Author Comment (AC2)

**Reply on Comment (Anonymous Referee #1)**

We are grateful to the anonymous referee for their careful reading of our manuscript and the constructive suggestions.

Below, each comment from the referee is *in italicized font*, and our responses in red normal font. Any new or added text in the manuscript is underlined in red, deleted text is with  in red, and these changes will be incorporated into the next revision.

All the line numbers in this reply refer to the original version of EGUsphere Manuscript ID: **egusphere-2025-2004**

*Here is my review for "Detecting irrigation signals from SMAP L3 and L4 soil moisture: A case study in California's Central Valley" submitted to Hydrology and Earth System Sciences. The paper aims to detect irrigation signals by comparing SMAP Level 3 and Level 4 soil moisture products, which is demonstrated in California's Central Valley. The results highlight the potential of satellite-based observations to identify irrigation effects. However, several aspects of the methodology, data usage, and interpretation require clarification and improvement to strengthen the study's scientific rigor and clarity. My recommendation is major revision to address the concerns outlined below. Below, I provide general comments on the manuscript's overall contribution, followed by specific comments to guide the authors in revising their work.*

**Reply:** We appreciate your time and efforts devoted to reviewing our work. We have carefully considered your comments and will revise the manuscript accordingly as outlined in the following.

**General Comments:**

*The authors note that SMAP L4 assimilation is based on brightness temperature anomalies, thereby excluding persistent irrigation effects embedded in the*

*climatological mean. However, in practice, irrigation is not always applied consistently during soil moisture deficits; its timing can be irregular and largely influences soil moisture anomalies rather than the climatology. Moreover, this assumption neglects potential non-stationary changes in irrigation practices driven by agricultural policies. I recommend that the authors assess whether the identified irrigation dates are realistic and consider conducting a synthetic experiment to demonstrate that SMAP L4 indeed lacks any irrigation-related signal.*

**Reply:** Thank you very much for your helpful comments and suggestions.

We agree that irrigation is not always applied consistently during periods of soil moisture deficit. The irregular irrigation application can influence soil moisture anomalies rather than long-term climatology, meaning that in some scenarios the SMAP L4 may indeed still contain irrigation-related signals or effects. **As noted in our discussion (Section 5.1), when there is a non-negligible amount of precipitation during the irrigation season, our approach does not perform well**. This description is consistent with your concerns regarding the anomalies versus climatology. It is challenging to draw a clear line between irrigation signals that appear more random (randomly timed from year to year) and those that behave more like a climatology (the similar irrigation patterns every year).

In contrast, in areas where there is little to no rainfall during the irrigation season (as in the California Central Valley and the Snake River Basin), during the cropping season there is only negligible rain and irrigation is required every year to maintain soil moisture for crop growth. In these regions, irrigation is applied in a consistent pattern year after year. As a result, the added water becomes **part of the expected climatology rather than an anomaly**, **meaning it is not assimilated in the SMAP L4 product**. Under these conditions, **our approach works well, since the irrigation signal remains distinct and detectable**.

We also appreciate the suggestion to conduct a synthetic experiment that examines irrigation-related effects on the SMAP L4 product. However, at our current capacity, it is very challenging to isolate an independent synthetic experiment from the SMAP

production workflow to directly demonstrate the absence of irrigation signals in the SMAP L4 soil moisture. While the SMAP L4 Handbook and User Guide[2] provide detailed descriptions of the data assimilation process, very little insight is offered into how irrigation effects are handled. Nonetheless, we believe it is crucial to show that **assimilating anomalies does not cause SMAP L4 to carry irrigation effects**. To address your concern, we plan to **include a synthetic experiment** in the next version of the manuscript, as outlined below:

**Step #1:** Baseline synthetic experiment setup. The synthetic experiment will be based on the surface soil water mass balance equation, as Eq. (R1) shows:

$$\Delta z \frac{d\theta(t)}{dt} = P(t) - ET(t) - Q(t) = P(t) - L(t) \qquad (R1)$$

where $\Delta z$ is the depth of soil control volume, $\theta$ is the volumetric soil moisture, $P$ is precipitation, $ET$ is evapotranspiration, $Q$ is drainage and runoff, and $L = ET + Q$ is the "loss function". On the grid cells where irrigation is needed, surface runoff is negligible, and runoff term can be omitted, simplifying the loss function as $L = ET$. Following McColl et al. (2019), the precipitation will be generated using Eq. (R2) [2].

$$\bar{P}(t) = \overline{P_0} + A_{\bar{P}} \sin\left(\frac{2\pi t}{365}\right) \qquad (R2)$$

where $t$ is the day, $\bar{P}(t)$ is the annually mean daily precipitation, and $A_{\bar{P}}$ is the amplitude of the seasonal cycle.

The loss function $L$ will be generated by Eq. (R3) as follows:

$$L = \begin{cases} E_{max}, s \geq s_* \\ \frac{s-s_w}{s_*-s_w} E_{max}, s_w \leq s \leq s_* \\ \frac{K_s\left[e^{\beta(s-s_{fc})}-1\right]}{e^{\beta(1-s_{fc})}-1}, s \leq s_* \end{cases} \qquad (R3)$$

where $s = \theta/n$ is soil saturation, $n$ is the soil porosity, $s_w$ is wilting point, $s_*$ is a critical saturation at which the transition from stage-I to stage-II $ET$ occurs, $s_{fc}$ is field capacity, $K_s$ is the saturated hydraulic conductivity, and $\beta$ is an empirical parameter controlling the shape of the drainage curve.

Eq. (R1) above indicates the baseline experiment where no irrigation will be incorporated into soil moisture.

**Step #2:** Irrigation synthetic experiment setup. For irrigated grid cells, irrigation is

introduced as an additional water input term alongside precipitation as shown in Eq. (R4):

$$\Delta z \frac{d\theta(t)}{dt} = P(t) + I(t) - L(t) \tag{R4}$$

where $I(t)$ is irrigation water input.

We will then introduce two kinds of irrigation scenarios:

Scenario (1): a continuous irrigation water input (e.g., fixed irrigation interval for each year) during the typical irrigation season. This yields a consistent soil moisture time series with irrigation. Detailed information on the settings of irrigation will be provided in the next version of the manuscript.

Scenario (2): a scenario where the irrigation water input is highly variable across years (e.g., highly variable irrigation interval). This scenario helps us understand how anomaly will be in such regions with variable or intermittent irrigation practices.

**Step #3:** Anomaly Comparison: We will run these simulations over multiple years and compute the soil moisture anomalies (e.g., deviations from the long-term mean seasonal cycle) for both the non-irrigation and irrigation scenarios. We will then compare the anomaly time series between the two scenarios. If the anomalies from the irrigation scenario closely resemble those from the control scenario, it indicates that the presence of irrigation does not significantly alter the anomaly patterns. In other words, assimilating anomaly data (as is done in SMAP L4) would not introduce irrigation signals into the soil moisture product.

We hope that this synthetic experiment will address your concern. We will include the supplementary information of this synthetic experiment in our revised manuscript, and we will also this clarification in the revised version:

Line 148: Based on the above explanation, in regions where irrigation is irregular or subject to significant changes (for example, due to evolving agricultural policies), these non-stationary irrigation practices can alter the SMAP L4 climatology baseline, introducing irrigation-related anomalies into the assimilated data. In other words, under highly irregular irrigation regimes, irrigation effects could indeed be present in the

SMAP L4 product, which would limit the effectiveness of the proposed approach.

We thank you once again for these insightful comments and suggestions. They have helped us improve the rigor of our work, and we hope that the revisions will address the concerns raised.

[1] *U.S. Department of Agriculture, National Agricultural Statistics Service, 2024. Irrigation and Water Management Survey. https://www.nass.usda.gov/Surveys/Guide_to_NASS_Surveys/Farm_and_Ranch_Irrigation/*

[2] *K. A. McColl, Q. He, H. Lu, and D. Entekhabi, 2019. Short-Term and Long-Term Surface Soil Moisture Memory Time Scales Are Spatially Anticorrelated at Global Scales. J. Hydrometeor., 20, 1165–1182, doi: 10.1175/JHM-D-18-0141.1.*

[3] *R.H. Reichle, G. De Lannoy, R. Koster, W. Crow, J. Kimball, Q. Liu, M. Bechtold, 2022. SMAP L4 global 3-hourly 9 km EASE-Grid surface and root zone soil moisture geophysical data, Version 7. doi:10.5067/EVKPQZ4AFC4D*

**Specific Comments #1:** *Line 145: Limitation of SMAP L4 Assumptions*

*The manuscript states that SMAP L4 excludes irrigation signals in croplands with continuous irrigation. This assumption significantly limits the method's applicability, as it may not account for regions with variable or intermittent irrigation practices. The authors should clarify the conditions under which this assumption holds and discuss its implications for the method's generalizability.*

**Reply:** Thanks for your helpful comments and we think this comment expressing the same thing as the General Comments.

Please refer to the Reply to the General Comments above.

**Specific Comments #2:** *Section 2.1.3: Outdated Irrigated Area Map*

*The manuscript relies on a 2005 Global Map of Irrigated Areas (GMIA) dataset, despite using SMAP data from 2016–2022. Given potential changes in irrigation extent over this 15-year period due to agricultural expansion, urbanization, or policy shifts, the use of an outdated map introduces uncertainty. The authors should justify this choice or consider incorporating a time-series irrigated area dataset to align with the SMAP data*

*period. A discussion of how changes in irrigation patterns might affect the results is also warranted.*

**Reply:** Thanks for your valuable suggestions.

We applied the Global Map of Irrigated Area (GMIA) Version 5 as it remains **the latest publicly available and extensively validated global irrigated-area dataset**. GMIA Version 5 has undergone extensive validation by Siebert et al.[1] and is recognized as one of the most reliable global irrigation-related datasets currently available[2]. It still serves as **a foundational dataset input for irrigation representations in the global hydrological models**, including PCR-GLOBWB, WaterGAP, and H08. On the other hand, GMIA's native 5 arcmin resolution closely matches the EASEv2.0 M09 grid used by SMAP ($\approx 9\,$km at the equator), which minimizes uncertainties associated with resampling and interpolation.

However, we acknowledge that newer irrigated area maps have since been developed (e.g., MIRCA-OS [3], MapSPAM [4], and Nagaraj et al. 2021's global irrigation dataset for 2001–2015 [5]). We are currently reviewing these alternative datasets and intend to include an expanded discussion in our supplementary materials regarding the potential implications that the temporal mismatch between the GMIA dataset (2005) and the SMAP data period (2016–2022) may have on our results.

Accordingly, we propose to include the following text in the next version of the manuscript:

Line 158: The GMIA is a static dataset that represents the percentage of irrigated land relative to the total grid cell area around 2005 (Siebert et al., 2013). It is still recognized as one of the most reliable global irrigation-related datasets currently available.

Line 378: since the GMIA was estimated based on the global irrigated land data in 2005 (Siebert et al., 2013), any subsequent changes are not captured in the GMIA. The temporal mismatch between the GMIA dataset and the SMAP data period is further discussed in the Supplementary Information Section S10.

Supplementary Information Section S10 (new): We will add a discussion of the newer irrigated area datasets and how changes in irrigated area maps may affect our findings.

[1] *S. Siebert, V. Henrich, K. Frenken, J. Burke, 2013. Global map of irrigation areas version 5.*

[2] *S. McDermid, M. Nocco, P. Lawston et al., 2023. Irrigation in the Earth system. Nature Review earth &environment. (4): 435-453*

[3] *E.A. Kebede, K.O. Oluoch, S. Siebert et al., 2025. A global open-source dataset of monthly irrigated and rainfed cropped areas (MIRCA-OS) for the 21st century. Scientific Data 12, 208*

[4] *Q. Yu, L. You, U. Wood-Sichra et al., 2020. A cultivated planet in 2010 – Part 2: The global gridded agricultural-production maps, Earth Syst. Sci. Data, 12, 3545–3572*

[5] *D. Nagaraj, E. Proust, A. Todeschini, M.C. Rulli et al., 2021. A new dataset of global irrigation areas from 2001 to 2015. Advances in Water Resources. (152): 103910*

**Specific Comments #3:** *Equation (2): Handling Precipitation Bias and Crop Rotation*

*Equation (2) defines the irrigation signal as the difference in mean soil moisture difference (MD) between cropping and non-cropping seasons, based on SMAP L3_E and L4 data in irrigated grid cells. However, line 270 notes that high MD values in the non-cropping season may result from precipitation, which could confound irrigation signals. The authors should address how precipitation biases and rain/no-rain errors are considered during the cropping season. Additionally, the method assumes distinct cropping and non-cropping seasons, which may not apply to regions with year-round crop rotation. The authors should discuss the method's applicability to such regions and propose potential adaptations.*

**Reply:** Thank you for your helpful comments and suggestions.

Our study defines the irrigation signal (*IS*) as the difference between the *MD* values for the cropping and non-cropping seasons, where for each season the *MD* is defined as the inter-product difference (i.e., *MD* = SMAP L3_E minus L4 soil moisture). By design, this approach **allows for systematic biases (differences in rainfall responses) between SMAP L3_E and L4 products, while only requiring that outside irrigated areas such systematic biases remain consistent across the cropping and non-cropping seasons**. Therefore, before detecting irrigation signals within irrigated grid cells, our method requires validating whether this assumption (the consistency of

SMAP L3_E and L4 systematic biases between cropping and non-cropping seasons) holds true in non-irrigated grid cells of the study area. If most non-irrigated grid cells show inconsistent biases between cropping and non-cropping seasons, any detected *IS*, no matter how substantial, would not be reliable. Conversely, if the consistency assumption holds, we can expect that a high *MD* during the non-cropping season would correspond to a similarly high (in non-irrigated grid cells, reflecting the baseline systematic bias) or higher (in irrigated grid cells, baseline bias combined with irrigation effects) *MD* during the cropping season. Under such circumstances, irrigation signals can still be detected. In summary, the proposed definition method of *IS* effectively minimizes potential confounding from precipitation when detecting irrigation signals using SMAP L3_E and L4 products.

Our method implicitly assumes distinct cropping and non-cropping seasons, which could be identified based on local cropping calendars. At present, the proposed method is generally not applicable to regions with multiple cropping seasons or year-round irrigation because the baseline *MD* between SMAP L3_E and L4 in a non-cropping season is essential for calculating the irrigation signals. Figure R1 provides two representative grid cells in the Lower Mississippi flood basin (a region typically having two cropping seasons, with winter-spring irrigation) as a demonstration. It can be observed that, although SMAP L3 soil moisture is significantly higher than L4 during June to November (the summer-autumn cropping season), it approaches saturation during December to May (winter-spring cropping season), while SMAP L4 remains relatively lower due to the potential removal of irrigation effects. Although this observation partially supports our assumption that SMAP L3 captures irrigation signals while SMAP L4 largely excludes them (SMAP L4 in generally lower than L3 throughout the whole year), our current method does not enable a reliable estimation of irrigation signals in Lower Mississippi flood basin, since it is a typical region with two cropping seasons, where our method is not applicable yet.

[Figure]

**Figure R1: SMAP surface soil moisture from (red dots) L3_E and (blue dots) L4 for two representative grid cells in the Lower Mississippi flood basin.**

**Note that the manuscript already includes discussion of this limitation**. E.g., in Line 410 we caution that our method can effectively capture irrigation signals only in regions with limited cropping-season precipitation (e.g., (semi-)arid, (semi-)Mediterranean climatic regions), which usually feature single-season cropping or rain-fed crops in the winter. In these regions, we can reliably identify a baseline *MD* in the non-cropping season unaffected by irrigation, thereby detecting irrigation signals during the cropping season. In the revised version, we will strengthen the discussion of the applicability of our method. In future research, we would like to explore alternative approaches to determine baseline *MD* without irrigation. For instance, using *MD* values from adjacent non-irrigated grid cells as a baseline reference, and reassess irrigation signals accordingly.

We appreciate your constructive comments, which significantly enhance the scientific rigor of our manuscript. In the next version, we will revise the relevant text as follows: Line 244: Note that *IS* represents irrigation intensity rather than the absolute irrigation water use amount; this distinction is further elaborated in Section 5. Additionally, due to the requirement to differentiate between cropping and non-cropping seasons, our method is currently not applicable to regions with multiple cropping cycles or year-round irrigation.

**Specific Comments #4:** *Line 255 and Table 1: Inconsistent Threshold for Non-Irrigated Grid Cells*

*Line 255 states that non-irrigated grid cells have an irrigated area fraction below 0.1%, but Table 1 reports a threshold of 0.11% (non-irrigated grid cell b).*

**Reply:** Thanks for your careful revision and comments.

We re-checked the irrigated area fraction of grid cell (b), which is 0.011% (and not 0.11%). We apologize for the typo and will correct it in the next version as follows:

Table 1, Irrigated area fraction of Grid cell (b):  0.01%

**Specific Comments #5:** *Figures 2(d) and 3(c): Bias in Soil Moisture Differences*

*Figures 2(d) and 3(c) show a higher mean difference between SMAP L3 and L4 in non-irrigated grid cell E (0.0471 m³/m³) compared to irrigated grid cell C (0.0354 m³/m³). This bias may stem from using the outdated 2005 GMIA dataset, as irrigation patterns may have changed. The authors should investigate whether this discrepancy reflects changes in irrigation extent or other factors (e.g., soil properties, land cover) and discuss their findings.*

**Reply:** Thank you for your valuable comments and suggestions.

We would like to firstly clarify that **the comparison of *MD* between cropping and non-cropping seasons should only be conducted independently within each grid cell individually**, as defined in Eq. (2) of the original manuscript (i.e., *MD* for the cropping season compared to *MD* for the non-cropping season of a specific grid cell). This is because different grid cells have distinct soil characteristics, land cover, and other factors, the systematic bias baselines of *MD* vary among grid cells. Therefore, careful caution and test are needed when directly comparing *MD* values between different grid cells. This is also the reason why, as discussed in our reply to Specific Comment **#3**, we are still trying to test the potential use of *MD* from neighboring non-irrigated grid cells as a uniform baseline to quantify multi-seasonal irrigation effects. Specifically, in the example you mentioned, the non-irrigated grid cell e in Figure 2(d)

indeed shows a higher *MD* than the irrigated grid cell C in Figure 3(c). However, the baseline *MD* of cell e is much higher than that of cell C (0.0717 m³/m³ vs. -0.0105 m³/m³), indicating substantial differences in systematic bias between SMAP L3 and L4 at two grid cells. Hence, from a methodological perspective, these two *MD* values should not be directly compared.

Additionally, we understand that the use of GMIA dataset based on year 2005 may introduce uncertainties into the validation process, as you have pointed out in Specific Comment **#2**. As mentioned in our response to Specific Comment #2, we will examine a newer irrigated area map to explore the potential impacts of the GMIA dataset and provide additional analysis in the supplementary information of the next version of manuscript.

To clarify the applicability of the proposed approach, we will add the following revised text in the next manuscript version:

Line 234: Note that the above analyses should be conducted independently for each grid cell; *MD* values from different grid cells cannot be cross compared.

**Specific Comments #6:** *Line 320: Error in Supplementary Figure Reference*

*The reference to "Supplementary Information Figure S5-3" appears incorrect. Please verify and correct the figure number.*

**Reply:** Thank you very much for your careful revision.

We have checked the figure number for this reference, and the correct figure number should be "Figure S5-4" here. We apologize for the typo error here and all figure numbers have been re-checked again.

The figure number will be revised in the next version:

Line 320: Supplementary Information Figure S5-3 S5-4

**Specific Comments #7:** *Figures 5 and S-4: Interpretation of $R_{non}$ Performance*

*The manuscript reports that 72.7% of $R_{non}$ values fall below the consistency threshold, suggesting limitations in the method's performance for SMAP L4. The authors attribute this to the absence of saturated soil moisture values in L4 compared to L3_E during the non-cropping season. However, SMAP L3_E's sensitivity to surface wetting (e.g., post-rainfall or standing water) may lead to unrealistic retrievals. The authors should clarify whether these values reflect true soil saturation or retrieval error.*

**Reply:** Thanks for your insightful comments, and we appreciate the opportunity to clarify this point in the next version.

Our analysis indicates that these saturated SMAP L3_E soil moisture values are likely retrieval artifacts caused by standing water after heavy rainfall, rather than true soil saturation. Specifically, we observed that the saturation points in SMAP L3_E occur during the non-cropping season (winter-spring) in SV, following periods of intense precipitation. This timing suggests that the SMAP L3_E retrieval is responding to temporary surface wetting or ponding. Previous studies[1] have reported that even if a small fraction of a radiometer pixel (5% or more) is covered by standing water, SMAP L3_E algorithm can overestimate the soil moisture, sometimes by ~0.2 m³/m³, leading to unrealistically high retrievals. In our study case, SV experienced consecutive heavy rain events in winter-spring, which likely created standing water that the L3_E interpreted as saturated soil. In contrast, the SMAP L4 product uses a land surface model and assimilates brightness temperature anomalies (rather than directly using the radiometer's absolute soil moisture retrievals). This approach makes L4 less sensitive to transient standing water effects, so it shows high soil moisture but does not reach the saturation that L3_E does.

It should be noted that these potentially saturated points in SMAP L3_E limit the area where our method can be applied, they do not compromise our irrigation analysis results in areas where our method can be applied. We conducted a climatological consistency test between SMAP L3_E and L4, and the SV grid cells with low $R_{non}$ (inconsistent

wet-season behavior) failed that test. Consequently, even though those grid cells might contain irrigated areas, they were excluded from subsequent analyses. In other words, they were flagged as having inconsistent climatology and were not used when mapping irrigation signals.

We will revise the following description to clarify this issue in the next version:

Line 323: These saturated L3_E values are likely to be retrieval errors caused by surface wetting (e.g., standing water following heavy rainfalls) (Ye et al., 2015). SMAP L4's assimilation approach avoids these spurious saturation signals.

[1] *N. Ye, J.P. Walker, J. Guerschman et al., 2015. Standing water effect on soil moisture retrieval from L-band passive microwave observations. Remote Sensing of Environment. 169 (2015): 232-242*

**Specific Comments #8:** *Figure 7: Interpretation of Irrigation Signal (IS)*

*The statement in line 245 that "a larger IS value indicates higher irrigation intensity" implies that any positive IS value signals irrigation activity. However, Figure 7 shows biases in some regions (e.g., region (i)), which may undermine this interpretation. The authors should clarify whether positive IS values consistently indicate irrigation or if biases (e.g., from precipitation or land cover changes) could lead to false positives. A sensitivity analysis of IS values would help address this concern.*

**Reply:** Thanks for your professional comments.

We understand that our original statement in Line 245 ("A larger *IS* value indicates higher irrigation intensity") may appear overly definitive, particularly given that Figure 7 reveals some biases in certain regions, and some could result from the confounding factors.

In response, we will revise the description accordingly in the next manuscript version as follows:

Line 245:  Ideally, a larger *IS* value would be associated with higher irrigation intensity; however, uncertainties caused by other factors (e.g., precipitation events, land cover changes)

could introduce biases.

Additionally, we will add a sensitivity analysis of *IS* values (spatial map) in the supplementary information to explicitly address this concern.

**Specific Comments #9:** *Figure 8:*

*The validation of the estimated irrigation signal, which relies on comparisons with the GMIA and irrigation water use datasets, is inadequate. Using GMIA as both an input to the method and a validation dataset introduces circularity, undermining the independence of the validation process. Additionally, the manuscript reports a spatial correlation of 0.5 with the ZL21 map, which is relatively low and raises questions about the method's accuracy. The authors should clarify the significance of this correlation value and consider incorporating independent validation datasets (e.g., in-situ irrigation records) to strengthen the evaluation of the irrigation signal estimates.*

**Reply:** Thanks for your valuable comments.

We would like to clarify that our method uses the GMIA dataset in two independent steps. First (Section 3.1), GMIA is used solely to sample irrigated and non-irrigated grid cells to evaluate whether SMAP L3_E and L4 products are climatologically consistent within the study region; beyond this sampling, GMIA does not inform any further analysis. Second (Section 3.2), GMIA is applied to validate our estimated *IS* map. These two steps employ different procedures, and the first step does not provide any information related to GMIA to the second step, we believe the risk of circular validation is minimized.

The relatively low correlation coefficient is primarily attributed to two possible factors: (1) Different soil-moisture depths: ZL21 is based on rootzone soil moisture, whereas our *IS* method relies solely on surface soil moisture from SMAP L3_E and L4. In Central Valley, subsurface irrigation practices cannot be fully captured by surface-only observations. (2) Resolution mismatch: ZL21's native resolution is 1 km, which we

resampled to 9 km for direct comparison with the *IS* map, potentially diluting finer-scale irrigation patterns. Accordingly, we will add the following text to the next version of the manuscript:

Line 365: The relatively lower correlation with the ZL21 dataset likely arises since ZL21 utilizes rootzone soil moisture to estimate irrigation, which the proposed IS map cannot capture by the surface retrievals from SMAP products.

Regarding in-situ irrigation observations, we must acknowledge that obtaining large-scale, continuous, reliable, publicly available in-situ irrigation records is very challenging. This limitation motivated our inclusion of the ZL21 map alongside GMIA for validation. On the other hand, even if farm-level irrigation withdrawal data were accessible, the mismatch between small-scale irrigation practices and SMAP's 9 km spatial resolution would likely introduce validation errors.

Once again, we greatly appreciate your helpful comments, which have significantly enhanced the scientific rigor of our manuscript.

---

## Author Comment (AC3)

**Reply on Comment (Anonymous Referee #2)**

We thank the anonymous referee for the valuable suggestions on our manuscript.

As detailed below, the referee's comments are *in italicized font*, and our responses in red normal font. Any new or added text in the manuscript is underlined in red, deleted text is with  in red, and these changes will be incorporated into the next revision.

All the line numbers in this reply refer to the original version of EGUsphere Manuscript ID: **egusphere-2025-2004**

**Referee #2's General Comments:**

**Reply:** We appreciate the referee's time and efforts devoted to reviewing our work. Since the general comment raises multiple major issues, we will **separate it into individual points** and provide the point-to-point responses below.

**General Comments, Point #1:**

*The authors propose an approach for detecting irrigation signals through discrepancies between SMAP L3 (potentially able to track irrigation) and L4 data (unable to track irrigation). The strength of the methodology consists in solely relying on data without the need of any modeling. Analyses are performed rigorously and results are clearly presented.*

**Reply:** Thank you for your revision and valuable comments, and we are grateful for the positive remarks regarding the clarity of our methodology and results.

**General Comments, Point #2:**

*Nevertheless, the study appears as a bit out-of-date with respect to the current status of*

*the art in terms of irrigation monitoring through satellite data. In fact, in light of several well-established methodologies for detecting in space and time irrigation events and for estimating irrigation water use, with some of them even close to facing operational implementation, the current study appears limited in its scope.*

**Reply:** We appreciate the referee's valuable opinions, although we perceive that some of the comments may stem from different expectations regarding the study's scope and methodology. We agree that satellite-based irrigation monitoring is an active research area, with several methods that are even practically applicable. However, synergistically leveraging SMAP L3 and L4 for irrigation detecting purpose and in a purely data-driven manner, as in our study, provides a unique approach to resolve the common problems of inconsistent soil moisture climatology among different datasets, and efficiently reduce the complexity of model tuning, as detailed below:

- **Consistent soil moisture climatology:** Unlike methods that rely on comparing satellite data to independent models or reanalysis (e.g., using MERRA-2 as a non-irrigated baseline in Zaussinger et al's research[1]), our approach uses two products showing highly consistent soil moisture climatology. SMAP Level 3 is a direct satellite retrieval (which includes the real-worlf irrigation-induced soil moisture increases), while SMAP Level 4 is a model-assimilated product that assimilates only brightness temperature anomalies, containing no irrigation effects. Both products employ nearly identical radiative transfer algorithms, ensuring their soil moisture climatology is highly consistent. This consistency minimizes potential biases and false signals; a key novelty compared to prior studies that often struggled with climatological mismatches between different datasets.

- **Purely data-driven (no complex tuning or manual screening):** Our method does not require any additional model calibration or tuning beyond the standard SMAP processing. By taking the difference between SMAP Level 3 and Level 4, we detect irrigation signals directly from observations. This stands in contrast to approaches that integrate satellite data into hydrological models or require sophisticated adjustments.

Our SMAP Level 3 vs Level 4 differencing approach offers **a novel, simple way to identify irrigation signals**. By using two standard SMAP products, we ensure data consistency and avoid the biases that can arise in other techniques. We believe this is the first study to use SMAP L4's inherent "non-irrigation" baseline in tandem with L3_E for irrigation detection, and we will underscore this innovative aspect more clearly in the revised paper as follows:

Line 451: In this study, we proposed a method to detect irrigation signals directly from SMAP L3_E and L4 soil moisture products. This method requires minimum additional data or model tuning, yet preserves a consistent soil-moisture climatology between the satellite observations (SMAP L3_E) and the non-irrigated baseline (SMAP L4). To our knowledge, this is the first study to employ SMAP L3 and L4 synergistically for irrigation detection.

[1] *Zaussinger, F., Dorigo, W., Gruber, A., et al, 2019. Estimating irrigation water use over the contiguous United States by combining satellite and reanalysis soil moisture data. Hydrol. Earth Syst. Sci. 23, 897–923.*

**General Comments, Point #3:**

*In addition, the capability of SMAP retrievals in detecting irrigation in California has been already proved in previous studies (see, e.g., https://doi.org/10.1002/2017GL075733, https://doi.org/10.1016/j.hydroa.2023.100169).*

**Reply:** Thank you for pointing this out and recommended papers.

We also agree that the studies you mentioned have demonstrated the potential of SMAP series to detect irrigation signals in California's Central Valley (e.g., Lawston et al., 2017[1]; Soylu and Bras, 2024[2]). These works greatly inspired our research. **However, we would like to clarify that our approach is fundamentally different from these studies in two key ways: (1) we combine SMAP Level 3 and Level 4 datasets in our analysis, a combination not explored in prior studies, and (2) we extend the analysis from a single-pixel demonstration to a spatially continuous regional map of irrigation signals.**

Lawston et al. (2017) relied on the SMAP Level 3 product only, comparing one known irrigated pixel with an adjacent non-irrigated pixel and attributing the difference in their time series to irrigation effects[1]. This elegant experiment proved that the SMAP Level 3 dataset does contain irrigation effects. However, the method requires prior knowledge of which areas are irrigated and involves labor-intensive, pixel-by-pixel comparisons, making it difficult to scale up. Moreover, while it qualitatively shows the presence of an irrigation effect, using it for quantitative estimates would raise questions about whether the chosen "non-irrigated" baseline is comparable across different grid cells. Soylu and Bras (2024) analyzed a single grid cell in California using SMAP Level 2 observations in conjunction with a calibrated bucket-type hydrological model as a non-irrigation baseline[2]. By comparing the observed soil moisture against this modeled baseline, they estimated the irrigation amount. As the authors acknowledged, their approach likely overestimates irrigation amounts and necessitates substantial effort to calibrate the model in order to maintain a consistent soil moisture climatology with SMAP Level 2 product. In other words, their method, while innovative, relies on model tuning and site-specific calibration beyond the satellite dataset itself.

Inspired by these studies, we sought to advance the concept of SMAP-based irrigation detection in a simpler yet more expansive way. In our approach, we **synergistically leverage SMAP Level 3 and Level 4 products**. By differencing these two products, we obtain an "irrigation signal" map that represents the irrigation intensity, while **maintaining climatological consistency between the soil moisture with irrigation effects and without irrigation baseline**. This method avoids the need for any external model or additional calibration, without extra forcing data or hydrological models. Furthermore, compared to the above studies, our study **moves beyond single-pixel analysis to generate a spatially continuous *IS* map** for the California Central Valley. It does not require pre-selecting control pixels or tuning models for each location, which makes it more practical for large-scale estimation.

To address your concerns, we will add a new table (Table R1 as follows) that compares

our method with the above two studies, along with a new text based on our discussion above. We believe these clarifications emphasize the novelty and broader applicability of our work, and we thank you for giving us the opportunity to explain this issue.

**Table R1: Comparison between the proposed method and previous studies**

| | (Potentially) irrigated reference | Non-irrigated reference | Requirements | Outputs |
|---|---|---|---|---|
| Lawston et al. (2017) | SMAP L3 product | SMAP L3 product at a **nearby** grid cell **known** to be non-irrigated | The non-irrigated reference is in fact not irrigated | **Qualitative evidence** of irrigation effects in SMAP L3 product |
| Soylu and Bras (2024) | SMAP L2 product | **Non-irrigated model simulation** of the potentially irrigated grid cell | The bias between model and SMAP L2 is known | Estimate of **irrigation amount** at the potentially irrigated grid cell |
| This work | SMAP L3 product | SMAP L4 product | A nearby grid cells known to not be irrigated to test the soil moisture climatology | **Irrigation intensity map** |

[1] *Lawston, P.M., Santanello, J.A., Kumar, S.V., 2017. Irrigation Signals Detected From SMAP Soil Moisture Retrievals. Geophys. Res. Lett. 44.*

[2] *Soylu, M.E., Bras, R.L., 2024. Quantifying and valuing irrigation in energy and water limited agroecosystems. J. Hydrol. X 22, 100169.*

**General Comments, Point #4:**

*On top of this, in addition to limitations discussed by the authors, those linked to the mismatch between the spatial resolution of SMAP retrievals and the extent of irrigated areas elsewhere are not mentioned but represent a critical point in the irrigation detection domain (https://doi.org/10.1016/j.jag.2022.102979).*

**Reply:** Thanks for your helpful suggestions.

We agree with your comment about the spatial resolution mismatch as an important consideration. The SMAP ~9 km grid, although relatively fine for passive microwave observations, is still coarse for detailed irrigation monitoring, for which a resolution on the order of 100 m is often desirable. As reported by Zappa et al. (2022), coarse pixels attenuate irrigation signals and can lead to underestimation of irrigation water use; reliable detection generally requires that at least one-third of a pixel be irrigated [1]. Consequently, our SMAP-based method may overlook small or fragmented irrigation.

We will state this limitation explicitly in the next version of the manuscript:

Line 179: It is important to note, as previous studies reported (Zappa et al., 2022; Zaussinger et al., 2019), that the coarse spatial resolution of satellite soil-moisture pixels often weakens irrigation signals. Consequently, satellite-based detection is most dependable in large, contiguous, and intensively irrigated regions, whereas results for small or scattered irrigation patches should be interpreted with caution.

[1] *Zappa, L., Schlaffer, S., Brocca, L., et al., 2022. How accurately can we retrieve irrigation timing and water amounts from (satellite) soil moisture? Int. J. Appl. Earth Obs. Geoinformation 113, 102979.*

**General Comments, Point #5:**

*In my opinion, the limited scope of this paper with respect to the current status of knowledge does not incentivize its publication. The paper does not propose an irrigation quantification method (because of the limits in retrieving irrigation fluxes clearly explained by the authors) neither an irrigation mapping approach (as the a priori knowledge of irrigated and non-irrigated pixels is required). It could be seen as a method for detecting irrigation events but definitely an effort is required for highlighting advantages with respect to previous studies (e.g., https://doi.org/10.3390/rs15051449 or https://doi.org/10.3390/rs12091456, to cite a few).*

**Reply:** Thanks for your valuable comments. However, we respectfully disagree with your opinions and would like to clarify our position as below:

Firstly, the novelty and unique aspects of our study have been addressed in our response to the **General Comments, Point #2 & #3** above. While our method cannot explicitly quantify irrigation volume (due to the limitations in retrieving absolute irrigation fluxes, as we acknowledge in the manuscript), **it can be considered as a form of irrigation intensity mapping**. After confirming the consistent climatology of SMAP Level 3 and Level 4 products, **this intensity map can be interpreted as a map of relative irrigation intensity or area**. In the Results section of original manuscript, we showed that this map correlates reasonably with validation datasets: it aligns with the Global

Map of Irrigated Areas (GMIA) in terms of spatial extent and with the ZL21 irrigation water use map. Moreover, our method can capture interannual variability in irrigation signals (year-to-year changes), which static irrigation maps cannot provide.

Second, **we emphasize that different satellite missions offer different strengths**. We appreciate the two recommended references. Those studies have indeed achieved satisfying results at finer spatial scales. However, the **Sentinel satellites have unique advantages of very high spatial resolution, whereas SMAP provides more frequent revisits, and a consistent modeling framework**. We believe **the success of Sentinel-based approaches does not diminish the value of exploring irrigation monitoring with other satellite products like SMAP**. The diverse applications of different soil moisture satellite observations can enrich the technological landscape of satellite-based approaches for irrigation detection, which will ultimately enhance the capability and accuracy of space-based irrigation studies.

In conclusion, while we respect the referee's valuable comments and expertise in this field, we insist that our study offers a novel perspective by exploiting the potential of SMAP satellite in irrigation detection, and that our research provides a complementary reference which can add to the community's knowledge of irrigation monitoring from space.

**Specific Comments #1:**

*L 21: To what temporal resolution do the correlation coefficients refer?*

**Reply:** Thanks for your valuable comments.

The irrigation signal (*IS*) map in this study is natively computed **annually**: for each year we take the difference between the mean SMAP L3_E – L4 soil-moisture difference during the cropping season and that during the non-cropping season. Thus, the native temporal resolution of each *IS* map (and of any correlation derived from it) is one year.

For ease of presentation, however, Figure 7 shows the **multi-year average *IS* map for 2016–2020**. The correlation coefficients reported in the manuscript therefore refer to this five-year mean field; in other words, they represent the spatial correlation of the 2016-2020 average *IS* map with the validation.

We will revise the title of Figure 8 of the manuscript to make this part clearer as follows:

Line 356: Figure 8 displays the scatterplot and *R* values between the estimated average *IS* map for 2016–2020 and the irrigated area fraction from the GMIA (Fig. 8a) as well as the average irrigation water use estimations from the ZL21 map (Fig. 8b).

Line 365: Figure 8: Scatterplot of grid cell values in the *IS* map (averaging value for 2016–2020) compared with those from the GMIA and the ZL21 map in SJV.

**Specific Comments #2:**

*L 62-75: SM-based methodologies for retrieving irrigation information can be divided into two main categories, namely baseline approaches (as for instance https://doi.org/10.5194/hess-23-897-2019, https://doi.org/10.3390/rs13091727) or methodologies based on the soil water balance (e.g., https://doi.org/10.5194/essd-15-1555-2023). Note that such methodologies led to the development of satellite-based irrigation water use datasets (https://doi.org/10.5281/zenodo.8086046), also available for the US (https://doi.org/10.5281/zenodo.14988198).*

**Reply:** Thanks for your suggestions. We agree that soil moisture-based irrigation retrieval methods fall into two main categories (baseline vs. soil water balance approaches) and that our manuscript needs to acknowledge this. We respectfully note that while soil water mass balance approaches are highly valuable, a deep evaluation of those methods is beyond our current scope. **Our goal in this work is to demonstrate the potential of a baseline method**.

We propose to retain the original structure of the paragraph while making the additions outlined below. The revised paragraph first introduces the classification (baseline vs. mass balance), then discusses baseline methods (our focus). We are confident that this

addresses your concern without deviating from our paper's focus on the baseline approach (i.e., using differences between SMAP Level 3 and Level 4 products to detect irrigation). We will revise the manuscript as follows:

Line 59: Satellite soil moisture methods for retrieving irrigation information are generally divided into two categories: baseline approaches and soil-water mass-balance approaches. The latter estimate irrigation by closing the mass balance with satellite soil-moisture observations and other hydrological fluxes, and they typically require assimilating these soil moisture data into complex land-surface models (Dari et al., 2023). The present study concentrates on the baseline approach. The basic principle for this soil  baseline method is taking the difference between two soil moisture time series with irrigation effects (usually from satellite products) and without irrigation effects (usually from model simulations without considering irrigation events) (Brocca et al., 2018).

Line 62: The key to the soil moisture-based irrigation  baseline approach is to ensure that the time series with and without irrigation effects are climatologically consistent.

Thank you again for pointing this out, and it has helped us improve the manuscript's scholarly completeness.

**Specific Comments #3:**

*L 90-92: So why California only is mentioned in the title?*

**Reply:** The California Central Valley is **the primary focus of our study**, and the most thorough testing and validation are carried out there (see Sections 4.1–4.3). Results from other irrigated regions in the CONUS are included only to illustrate how the method performs outside the Central Valley and to highlight potential limitations; they do not alter the main conclusions drawn for the Central Valley.

As noted in Line 90-92 of the original manuscript: We also examined several heavily irrigated regions elsewhere in the contiguous United States (CONUS) to further

evaluate the performance and applicability of the proposed method.

Central Valley remains the core study area, and the paper's key findings and figures are based on that region; consequently, we referenced California alone in the title. If the reviewers or the editor feel a broader title would better reflect the supplementary analyses, we are willing to revise the title accordingly.

**Specific Comments #4:**

*L 170: performances of ZL21 should be reported to understand its reliability as a comparative dataset.*

**Reply:** We will add a description of the reliability of the ZL21 product in the next version of the manuscript as follows:

Line 170: Derived from model simulations that integrate remote sensing-based evapotranspiration with simulated root zone soil moisture, the ZL21 map offers high-resolution (1 km) monthly irrigation water use estimates for the CONUS over the period 2000–2020. Compared against state-level Farm and Ranch Irrigation Survey dataset, this product achieved $R^2$ values ranging from 0.74 to 0.84, demonstrating high accuracy at the state level.

**Specific Comments #5:**

*L 195: is flood irrigation an issue for detecting the irrigation signal?*

**Reply:** Thanks for your valuable comments.

We noted that the Sacramento Valley is dominated by flood irrigation systems, whereas the southern San Joaquin Valley primarily employs sprinkler irrigation to demonstrate that our method performs well under different irrigation practices, which has been confirmed by our results.

In principle, different irrigation practices can influence monitoring. Flood irrigation may create artificial standing water, and, as discussed in our Response to Referee #1,

Specific Comment #7, large areas of standing water can lead to an overestimation in SMAP Level 3 product and thus a more pronounced irrigation signal.

In summary, we believe **flood irrigation did not compromise our ability to reasonably detect irrigation signals using the SMAP Level 3 and Level 4 products in this study**.

**Specific Comments #6:**

*Figure 2: grid cell e) seems to show slightly different dynamics.*

**Reply:** Thanks for your careful revision and comments.

Compared with the other grid cells shown in Figure 2, grid cell e exhibits slightly different dynamics between SMAP L3_E and L4. We would like to first clarify that all non-irrigated grid cells were selected by a random-sampling procedure as described in Section 3.1 of the original manuscript. The feature of grid cell e is the higher SMAP L3_E soil-moisture values during the non-cropping season. As explained in our Response to **Referee #1, Specific Comment #7**, this behavior is likely caused by standing water after consecutive rainfall events, which can spuriously elevate SMAP L3_E.

Nevertheless, **this slightly different dynamics in grid cell e do not affect our overall assessment of climatological consistency** between SMAP L3_E and L4 over non-irrigated grid cells: their temporal variability and the *MD* between cropping and non-cropping seasons remain consistent, staying below 0.04 m³ m⁻³.

We thank you again for highlighting this detail.

**Specific Comments #7:**

*L 309: what is the entity of MD discrepancies?*

**Reply:** Thanks for your valuable comments.

As defined in Eq. (1) of the original manuscript: $MD = \frac{1}{n}\sum_{i=1}^{n}(\theta_{L3,i} - \theta_{L4,i})$, *MD*

represents the mean difference between SMAP L3_E and L4 soil moisture values over a given period.

In Line 309 we state: "At the irrigated grid cells (Figs. 4a–d), we generally observed higher *MD* values during the cropping season compared to those during non-cropping season ($p$-value < 0.05)", this means that, in irrigated grid cells, L3_E and L4 diverge more strongly during the cropping season than in the non-cropping season; in other words, SMAP L3_E shows higher soil-moisture levels than SMAP L4 during the cropping period but not because of the systematic errors. This systematic behavior constitutes the empirical basis for our subsequent irrigation-signal detection.

**Specific Comments #8:**

Figure 7: panel c), how il this map converted to m³/m³? Is porosity taken into account?

**Reply:** Thanks for your careful revision and comments.

Figure 7, Panel (c) is taken from the validation dataset: ZL21 map.

In this dataset, the authors use the ERA5-Land volumetric soil-moisture product, which is provided in units of m³ m⁻³. **Porosity is explicitly accounted for within the ERA5-Land land-surface model when it simulates soil-moisture dynamics**, so no additional conversion or porosity adjustment was required for their analysis.

**Specific Comments #9:**

*L 420-421: what about GLEAM, Sen-ET, ...*

**Reply:** Thanks for your valuable suggestions.

In the manuscript we noted the potential to refine our analysis by incorporating large-scale evapotranspiration products, while also mentioning the limited availability of independent observations. We appreciate your recommendation to consider GLEAM and Sen-ET. Both data sets indeed provide valuable ET estimates derived from satellite observations and/or reanalysis forcing, and we acknowledge their relevance to

irrigation studies. Although they remain, like most global ET products, indirectly constrained by remote-sensing or model inputs, they could nonetheless enhance the evaluation of water-balance components in future work.

We will explore integrating GLEAM and Sen-ET in follow-up studies and will reference these products in the revised manuscript.

Thank you again for your professional comments and suggestions.